# Improving Deep Learning for Accelerated MRI With Data Filtering

**Kang Lin**     **Anselm Krainovic**     **Kun Wang**     **Reinhard Heckel**
Department of Computer Engineering
Technical University of Munich
{`ka.lin, anselm.krainovic, kun2000.wang, reinhard.heckel`}@tum.de

## Abstract

Deep neural networks achieve state-of-the-art results for accelerated MRI reconstruction. Most research on deep learning based imaging focuses on improving neural network architectures trained and evaluated on fixed and homogeneous training and evaluation data. In this work, we investigate data curation strategies for improving MRI reconstruction. We assemble a large dataset of raw k-space data from 18 public sources consisting of 1.1M images and construct a diverse evaluation set comprising 48 test sets, capturing variations in anatomy, contrast, number of coils, and other key factors. We propose and study different data filtering strategies to enhance performance of current state-of-the-art neural networks for accelerated MRI reconstruction. Our experiments show that filtering the training data leads to consistent, albeit modest, performance gains. These performance gains are robust across different training set sizes and accelerations, and we find that filtering is particularly beneficial when the proportion of in-distribution data in the unfiltered training set is low.

## 1   Introduction

Deep neural networks achieve state-of-the-art results for accelerated MRI reconstruction [28]. While the majority of existing literature focuses on designing better neural network architectures for improving performance in accelerated MRI [15, 39, 10], research on effective dataset design for improving performance of neural networks for image reconstruction is limited. As a result, best practices for constructing datasets to train high-performing and robust models remain largely unclear.

In contrast, recent works in computer vision and natural language processing show that carefully curated training datasets can significantly boost model performance [11, 29, 12, 14, 19, 32]. For large foundation models, filtering an initial pool of web-scraped data for high-quality samples and training on this refined subset has led to substantial improvements across benchmarks [14, 12, 19].

We treat data as a fundamental part of model development, rather than a fixed resource, and demonstrate that curating training data through filtering candidate datasets can improve performance of existing state-of-the-art neural networks for accelerated MRI. For example, Figure 1 (left) shows for 8-fold accelerated MRI that a VarNet [39] (state-of-the-art for accelerated 2D MRI) trained on a smaller filtered dataset can provide a better reconstruction than the same model trained on the much larger unfiltered dataset. Our main contributions are as follows:

- We propose and investigate a variety of data filtering methods for improving training sets for deep learning based accelerated MRI. Similar to well-performing filtering approaches in the vision-language domain [12, 14], our best performing curation technique is based on retrieving images from the initial unfiltered training set that are similar to the validation data in terms of the DreamSim metric [13].

39th Conference on Neural Information Processing Systems (NeurIPS 2025) Track on Datasets and Benchmarks.

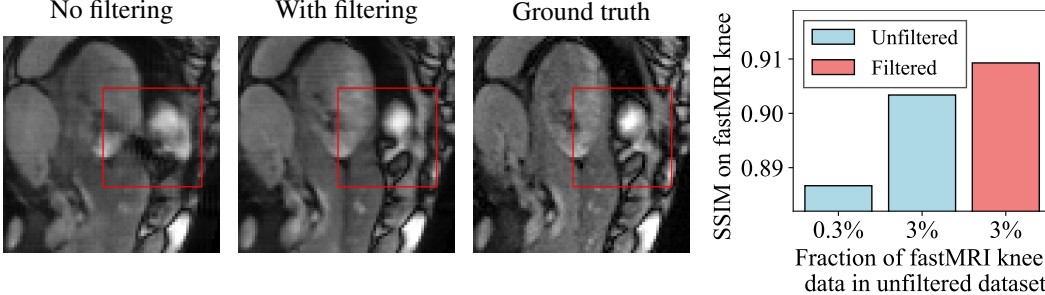

Figure 1: Performance of a VarNet [39] trained on an unfiltered dataset (120k slices) and on a filtered dataset (40k slices) for 8-fold accelerated MRI. **Left:** Cardiac MRI [44] reconstruction example showing that the VarNet trained on the filtered dataset yields a better reconstruction than the VarNet trained on the unfiltered dataset. **Right:** While a larger fraction of fastMRI knee data [47] in the training dataset results in a major performance boost on fastMRI knee test data, additionally filtering this dataset set results in further but smaller performance gains.

- We find that training on filtered datasets improves model performance compared to training on the unfiltered dataset, on both in-distribution and out-of-distribution data, with larger improvements on in-distribution data on average. However, we find that these performance gains are on average modest compared to starting with a better designed training set, such as one that includes more data from the distribution where high performance is desired. For example, as shown in Figure 1 (right), increasing the fraction of fastMRI knee data [47] in a fixed-size training set results in a major performance boost on fastMRI knee data. Applying filtering on top of this already improved dataset yields additional, but smaller gains.

- While the quantitative improvements from data filtering are modest, we find that they correspond to a visible reduction in small reconstruction artifacts and sharper details compared to training on the unfiltered dataset.

- We study how applying data filtering impacts reconstruction performance under different compositions and sizes of the unfiltered dataset, and across acceleration factors. We find that, compared to training on unfiltered data, filtering consistently leads to better performance when the unfiltered dataset contains a low fraction of in-distribution data. In our setups, the improvement from filtering is comparable to that of a 3-fold increase of unfiltered training data.

**Related work.** Several works show that data curation significantly impacts the performance of vision-language models (VLMs). For example, Schuhmann et al. [35] use a trained CLIP model [33] to curate a large-scale, open-source, multimodal dataset from web-scraped data. Models trained on this dataset achieve competitive results compared to state-of-the-art proprietary models. Similarly, Gadre et al. [14] investigate various data filtering approaches and propose a dataset to further improve VLM performance along with a benchmark to facilitate research in data curation. Fang et al. [12] further investigate training a model specifically for data filtering in the VLM context.

Similarly, in natural language processing, Li et al. [19] and Penedo et al. [32] show that carefully applying heuristics and machine learning models to filter large, uncurated text corpora leads to substantial gains in LLM performance.

Research on data curation for imaging is relatively limited. For natural image restoration, Yang et al. [45] and Li et al. [20] curate datasets from web-scraped images using heuristic filtering and show that training on their dataset yields slight improvements over existing datasets.

For accelerated MRI, several works introduce raw k-space datasets to facilitate machine learning research [47, 4, 23, 43, 44, 36], but do not study filtering or curation. Zbontar et al. [47] were the first to release a large, fully-sampled k-space dataset, which advanced the field. However, many subsequent works focus on improving neural networks [39, 28, 10], as opposed to the data. Lin and Heckel [22] emphasize the need for diverse k-space datasets to enhance robustness under distribution

Table 1: Fully-sampled k-space datasets used in this work. Scans containing multiple echoes, averages, or have a time component are separated as such and counted as separate volumes. Also, 3D MRI scans are converted to three individual volumes with a new slice direction depicting axial, sagittal, or coronal views.

| Dataset | Anatomy | View | Image contrast | Vendor | Magnet | Coils | Vol./Subj. | Slices |
|---|---|---|---|---|---|---|---|---|
| fastMRI knee [47] | knee | coronal | PD, PDFS | Siemens | 1.5T,3T | 15 | 1.2k/1.2k | 42k |
| fastMRI brain [47] | brain | axial | T1, T1POST, T2, FLAIR | Siemens | 1.5T, 3T | 4-20 | 6.4k/6.4k | 100k |
| CMRxRecon2023 [44] | heart | various | SSFP-Balanced | Siemens | 3T | 10 | 9.3k/300 | 58k |
| M4Raw [23] | brain | axial | T1, T2, FLAIR | XGY | 0.3T | 4 | 1.4k/183 | 25k |
| SKM-TEA [7] | knee | various | qDESS | GE | 3T | 8, 16 | 930/155 | 338k |
| AHEAD [3] | brain | various | MP2RAGE-ME | Philips | 7T | 32 | 1.1k/77 | 315k |
| fastMRI breast [36] | breast | various | VIBE | Siemens | 3T | 16 | 1.8k/300 | 499k |
| Lung 3D UTE [27] | lung | various | UTE | N/A | 3T | 23 | 69/23 | 18k |
| Chirp 3D [31] | brain | various | MPRAGE | Siemens | 3T | 17 | 6/1 | 1.4k |
| Extreme MRI [30] | lung, abdomen | various | SPGR, UTE | GE | 3T | 8, 12 | 6/2 | 1.7k |
| Fruits, Phantom [46] | N/A | various | MPRAGE | Siemens | 3T | 58, 64 | 6/2 | 1.9k |
| Heart T2-mapping [49] | heart | SAX | paper | Phillips | 3T | 32 | 44/12 | 1k |
| fastMRI prostate [43] | prostate | axial | T2 | Siemens | 3T | 10-30 | 312/312 | 9.5k |
| Stanford 2D [5] | various | various | various | GE | 3T | 3-32 | 89/89 | 2k |
| NYU data [15] | knee | various | PD, PDFS, T2FS | Siemens | 3T | 15 | 100/20 | 3.5k |
| M4Raw GRE [23] | brain | axial | GRE | XGY | 0.3T | 4 | 366/183 | 6.6k |
| SMURF [2] | knee, breast, abdomen | various | FSE, FatSat, WatSat, Dixon | Siemens | 3T | 10-20 | 113/11 | 1.3k |
| OCMR [4] | heart | various | SSFP | Siemens | 0.5T – 3T | 15-38 | 4.8k/165 | 1.3k |

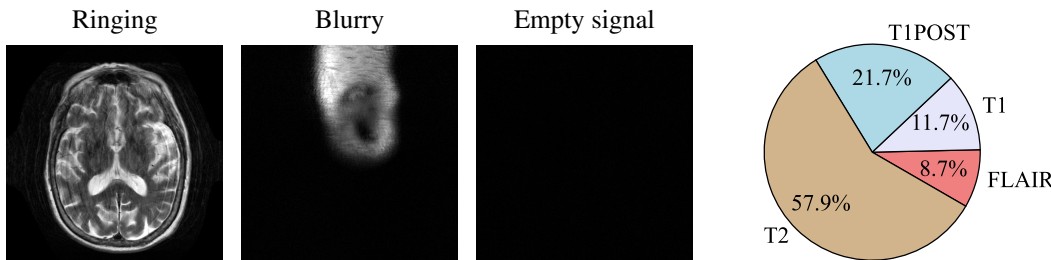

Figure 2: **Left:** Examples of low quality images within the fastMRI brain and knee test sets that we exclude from evaluation. **Right:** Skewed distribution of image contrasts within the fastMRI brain test set.

shifts but do not explore additional data curation strategies. In this work, we combine many existing open-source k-space datasets and explore data filtering for improving accelerated MRI.

## 2 Problem setup and data

In this section, we provide background on accelerated MRI, introduce the training data sources, the test data, as well as the models that we consider.

**Accelerated MRI.** We consider the problem of reconstructing a complex-valued image $\mathbf{x} \in \mathbb{C}^N$ based on undersampled measurements $\mathbf{y} \in \mathbb{C}^m$ in a multi-coil accelerated MRI setting. In this setup, $C$ receiver coils measure electromagnetic signals, and the measurements from the $i$-th coil are modeled as:

$$\mathbf{y}_i = \mathbf{MFS}_i\mathbf{x} + \mathbf{z}_i \in \mathbb{C}^m, \quad i = 1, \ldots, C, \tag{1}$$

where $\mathbf{S}_i$ is the spatial sensitivity map of the $i$-th coil, $\mathbf{F}$ denotes the 2D discrete Fourier transform, $\mathbf{M}$ is an undersampling mask that selects a subset of frequency components, and $\mathbf{z}_i$ is additive white Gaussian noise. The measurements $\mathbf{y}_i$ are known as k-space data. We focus on 2D MRI with Cartesian undersampling, where the central k-space region is fully sampled capturing 4%–8% of all k-space lines depending on the acceleration factor. The remaining lines are sampled equidistantly with a random offset from the start.

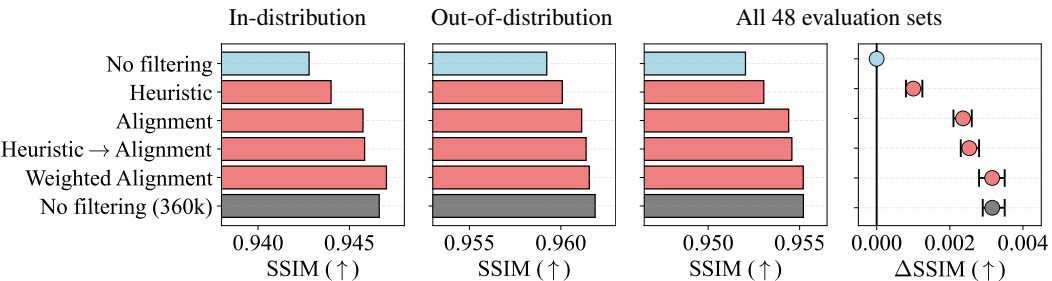

Figure 3: Data curation improves performance. For all investigated filtering methods, training on the filtered dataset improves performance over training on the unfiltered dataset (120k slices) on in-distribution and out-of-distribution evaluations. As a additional reference, the gray bar shows the performance obtained by training on a larger unfiltered dataset of 360k slices. We observe that using weighted alignment filtering matches the performance obtained by training on the larger unfiltered dataset. In the rightmost plot, we report the mean and 95% confidence intervals for performance gains over no filtering (at 120k slices), demonstrating that improvements are statistically significant.

**Datasets.** We utilize the data sources listed in Table 1. The first 12 sources serve as training data. Among these 12, the k-space data from fastMRI knee and brain [47], CMRxRecon2023 [44], and M4Raw [23] are acquired using 2D MRI sequences, whereas the other sources use 3D MRI sequences. Since we focus on models trained on 2D slices, we convert the 3D MRI k-space data into three separate volumes, each corresponding to axial, sagittal, or coronal views. This approach effectively increases the number of 2D slices available for training, yielding a total number of 1.1M slices for training.

**Evaluation.** We evaluate performance on accelerated 2D MRI. We only evaluate on data sources that are acquired with an actual 2D MRI sequence since this data enables realistic simulation of accelerated 2D MRI. Hence, in-distribution performance is evaluated on fastMRI knee, fastMRI brain, CMRxRecon2023, and M4Raw data. The last six data sources in Table 1 are used for out-of-distribution evaluation.

Many of the evaluation datasets are unbalanced in attributes such as contrast and magnetic field strength and often include lower-quality images with artifacts and noise. For example, Figure 2 illustrates that the fastMRI brain test set contains images with strong scanner artifacts and mainly T2-weighted images. To address this, we carefully curate our evaluation sets to ensure a more reliable assessment. First, we categorize the k-space data by data source, anatomy, anatomic view, contrast, number of coils, and magnetic field strength. From each group, we manually choose 5 to 24 images, ensuring diversity across subjects and slice coverage while excluding scanner artifacts. This selection process results in an evaluation suite of 48 curated test sets with 21 in-distribution test sets and 27 out-of-distribution test sets.

**Models.** Most of our experiments are with unrolled neural networks, specifically VarNets [39] with 80M parameters, since this type of network is the current state-of-the-art for accelerated 2D MRI reconstruction. In Appendix B, we also consider other neural networks trained end-to-end, specifically U-nets and Vision Transformers, and our conclusions for those are the same as for VarNets.

The VarNets take as input retrospectively undersampled measurements $\mathbf{y}$ and are trained with the objective to maximize SSIM between model output and the fully-sampled (i.e. $\mathbf{M}$ is identity) magnitude minimum variance unbiased estimator (MVUE) reconstructions. The sensitivity maps for computing the MVUE reconstruction are estimated with the BART toolbox [42]. The total training compute is chosen such that a model's performance saturates on a validation set that is curated in the same fashion as our evaluation suite.

Beyond networks trained end-to-end, in Section 3.6, we extend our results to diffusion-model based reconstruction methods.

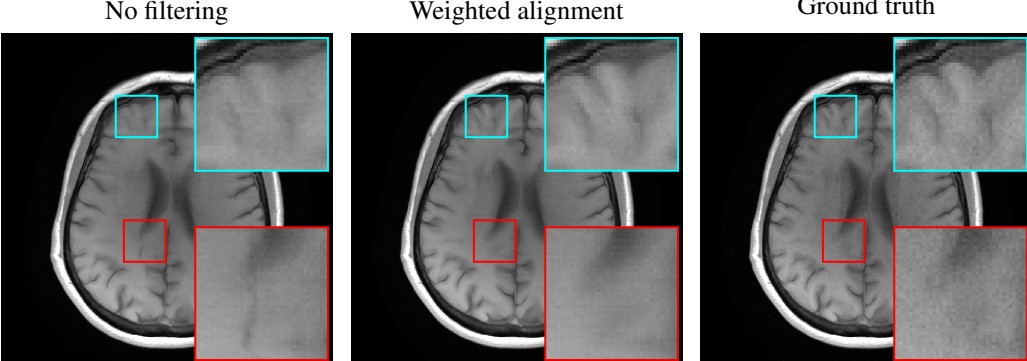

| No filtering | Weighted alignment | Ground truth |

Figure 4: Reconstruction examples at 4-fold acceleration showing that the reconstruction by the model trained without filtering contains small artifacts (red), which are substantially reduced by the model trained with weighted alignment filtering, while also providing sharper details (cyan).

# 3 Experiments

We consider two classes of filtering approaches: heuristic filtering (Section 3.1) and alignment-based filtering (Section 3.2). In Section 3.3, we compare the performance of the considered filtering approaches. Then, in Section 3.4, we analyze how performance behaves as we vary dataset size and the difficulty of the problem by changing the acceleration factor. Section 3.5 explores the relationship between performance and train-test similarity. Finally, in Section 3.6, we demonstrate that our findings for end-to-end models generalize to diffusion model-based reconstruction methods.

## 3.1 Heuristic filtering

MRI scans can contain images with visual degradation such as blurriness (for example, see Figure 2). Under the hypothesis that removing such low-quality data could help the neural network learn better image priors for reconstruction, we consider removing low-quality data from the training set.

To filter a dataset, we compute a score for each image within a dataset and keep an image when the score lies above a threshold. Our heuristic filter is a composition of the two heuristic filters below:

- **Energy filtering** identifies low-energy (i.e., dark) images. For a slice, we calculate its energy-ratio score $\frac{\max(\text{slice})}{\max(\text{volume})}$, where $\max(\text{slice})$ is the slice's maximum intensity and $\max(\text{volume})$ is the maximum intensity of the entire volume. A lower energy ratio corresponds to darker images. We keep slices with a ratio above $0.11$.

- **Edge-density filtering** identifies images that tend to be smooth or blurry. We first apply the Canny edge detector to compute the edges of an image. Then, the edge-density is calculated as the ratio of edge pixels to the total number of pixels in the image. We keep slices with a ratio above $0.017$.

Ablation studies on the threshold choices are provided in Figure 12 in the appendix.

## 3.2 Alignment-based filtering

Besides heuristic filtering methods, we consider alignment-based filtering. Alignment-based filtering has been successful for vision-language data [14, 12]. Gadre et al. [14] demonstrate that filtering data by retrieving data from the data pool that are similar to the benchmark data (in their case ImageNet) and training on this retrieved data improves performance on the benchmark compared to training on the entire data pool. We explore whether similar approaches can work for accelerated MRI and introduce two variants of alignment-based filtering: a default version which we call **alignment filtering** throughout, and an alternative version called **weighted alignment filtering**.

We leverage DreamSim [13], a perceptual image similarity metric that combines embeddings from CLIP, OpenCLIP, and DINO, fine-tuned on human judgments data. This metric aligns better with

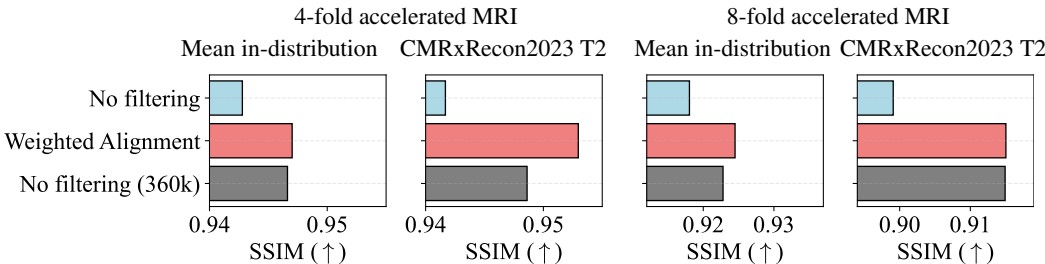

Figure 5: Compared to the mean performance gain obtained by filtering over no filtering, there exist data distributions on which filtering can significantly boost performance over no filtering (120k slices). As additional reference, the gray bar shows the performance gain obtained by training on a three times larger unfiltered dataset with 360k slices.

human image-similarity judgements than existing low-level metrics (such as PSNR, LPIPS) and semantic metrics (such as CLIP, DINO), and performs well on image retrieval tasks [41]. DreamSim computes the similarity between two images A and B as the cosine-similarity between the embeddings for images A and B.

To filter a dataset using DreamSim, we embed the magnitude of the fully-sampled MVUE reconstructions (i.e. the target images) in a dataset using the DreamSim model and do the same for each image in the validation set which is curated in the same fashion as the evaluation set. Then, for each embedding in the validation set we retrieve the images corresponding to the embeddings of the k-nearest neighbors within the dataset. Concretely, our default **alignment filtering** works as follows:

1. Preprocessing: Compute the magnitude image of the MVUE reconstruction for all slices in a dataset and in the validation set and normalize the magnitude images by the maximum magnitude pixel within their volume.

2. All magnitude images are then divided into non-overlapping image patches of size 128x128 pixels. These image-patches are embedded using the DreamSim model.

3. For each embedding, compute the cosine-similarity to the embedding belonging to the all zero image. Image patches with a similarity larger than 0.6 are discarded. This step removes image patches that are mostly empty.

4. To filter the dataset, retrieve for each embedding in the evaluation set, the images belonging to the k-nearest neighbors embeddings in the dataset.

5. Lastly, since the k-nearest neighbors of two different embeddings can contain the same image, remove all duplicates.

In our experiments, if not mentioned otherwise, we choose the number of nearest-neighbors such that 1/3 of the total number slices of the unfiltered dataset are retained. We ablate this choice on a random subset of the unfiltered data with 120k slices (see Figure 13 in the Appendix), and observed that retaining 20k to 40k slices yields similar best performance. While this choice works well in most of our experimental setups, it is not individually tuned for every setup.

**Weighted alignment filtering** omits Step 5 in the alignment filtering algorithm. This induces different sampling frequencies for images during training, i.e., images that are retrieved more often are also sampled more often during training. Instead of directly using the raw sampling frequencies obtained by omitting Step 5 in the alignment filtering algorithm, we take the square root of the raw sampling frequencies and use this output as sampling frequency of a slice. We observed that this approach yields slightly better performance than using the raw sampling frequencies.

**Deduplication.** Although our datasets do not contain slices with exact duplicates, near-duplicates occur due to very similar neighboring slices within a volume. This is often the case for the 3D MRI volumes considered here. Based on this observation, a potential caveat of alignment filtering is that the k-nearest neighbors of an embedding can contain many such near-duplicates which restricts the diversity of the retrieved dataset. To mitigate this problem, we remove near-duplicates within a volume before applying alignment filtering or weighted alignment filtering as follows: For each

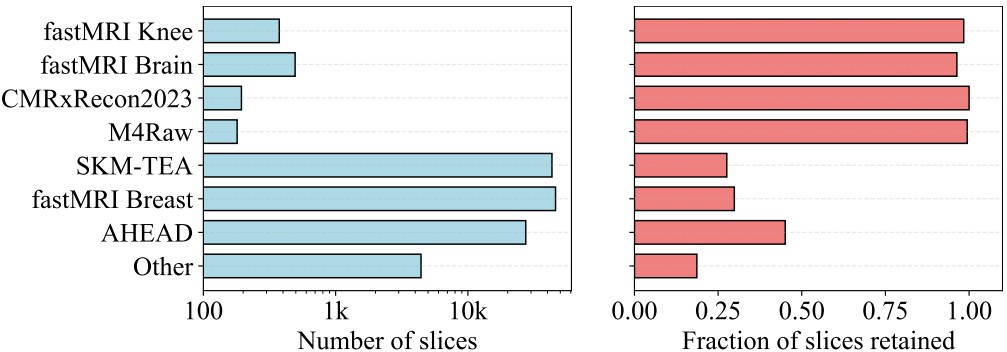

Figure 6: Number of samples for each data source in the unfiltered dataset with 120k slices in total (**left**) and fraction of samples remaining after alignment filtering 40k slices (**right**). After filtering, samples from in-distribution datasets (fastMRI knee, fastMRI brain, CMRxRecon, M4Raw) are kept almost completely.

embedding within a volume, we remove all other embeddings within the same volume if their similarity lies above a certain threshold, which we set to 0.9.

### 3.3 Main results

Figure 3 reports the performance of different filtering approaches. The unfiltered dataset considered for those experiments is a random subset of all volumes from the 12 training data sources totaling 120k slices. Section 3.4 reports results on the entire data pool containing 1.1M slices. Performance is reported as the average performance on all 48 test sets including in-distribution and out-of-distribution data. The main takeaways are as follows:

- All reported filtering methods improve over no filtering on both in-distribution data and out-of-distribution data.
- Alignment filtering provides better performance than heuristic filtering.
- Applying heuristic filtering first and then alignment filtering does not improve performance over only using alignment filtering.
- Weighted alignment filtering further improves performance and provides the best performance among our investigated filtering approaches.
- Overall, the mean performance gains from filtering are modest but statistically significant.

Given the modest average performance gains from filtering, a natural question is whether these improvements yield perceptible visual differences, especially when reconstructions are already mostly accurate, as in 4-fold acceleration. For example, a small numerical gain might only correspond to a slight change in brightness, which might not be perceptually significant. To investigate this, we assess how these gains appear in the test reconstructions. We find that often weighted alignment filtering reduces small reconstruction artifacts and yields sharper details compared to no filtering.

As shown in Figure 4, a model trained on the unfiltered dataset already produces an overall accurate reconstruction, but small artifacts remain. These artifacts are largely absent in the reconstructions produced by the model trained on the filtered dataset, while providing sharper details. More reconstructions are provided in the appendix in Figure 14 for 4-fold acceleration and Figure 15 for 8-fold acceleration.

Figure 5 illustrates that for certain data distributions, filtering can lead to a notable higher performance gain than the average gain. For example, on the T2-weighted cardiac images of the CMRxRecon2023 dataset [44], the performance gain (at around +0.01 SSIM) obtained by filtering is more than twice as high as the average performance gain at both 4-fold and 8-fold acceleration (a reconstruction example is shown in Figure 1). Figure 16 in the Appendix provides a detailed evaluation on all 48 test sets, where we observe that weighted alignment filtering improves on 46 out of those 48 test sets.

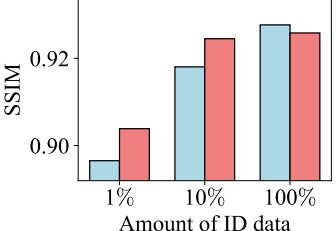 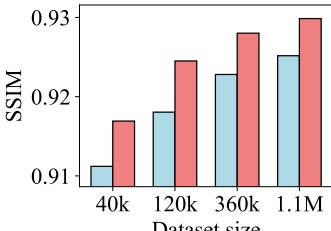 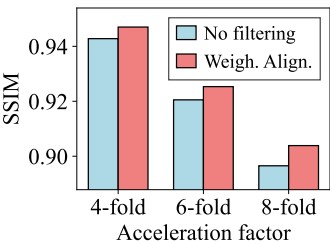

Figure 7: In-distribution (ID) performance at 8-fold acceleration as a function of the amount of in-distribution data in the unfiltered dataset. The unfiltered dataset size is fixed at 120k slices. Filtering improves performance when the fraction of in-distribution data is low. If the dataset is completely in-distribution, then filtering does not further improve performance.

Figure 8: In-distribution performance at 8-fold acceleration as a function of the amount of total data in the unfiltered dataset. The unfiltered datasets contain 10% in-distribution data. While filtering provides consistent gains across scales, it also significantly outperforms a randomly selected subset of the same size, as seen by comparing its performance against a same-sized unfiltered dataset.

Figure 9: In-distribution performance as a function of acceleration factor. The unfiltered dataset is fixed at 120k slices containing 1% in-distribution data. Weighted alignment filtering improves performance across accelerations. Slightly larger gains are observed at higher accelerations.

Appendix B contains additional details, results for other model architectures and explores how performance changes when models are fine-tuned on the validation set used for alignment filtering. Also for those setups, we observe that weighted alignment filtering yields performance gains over no filtering.

## 3.4 Ablation experiments

**Importance of in-distribution data in the unfiltered dataset.** Figure 6 shows the data distribution across different sources before and after alignment filtering. We observe that after filtering almost all data samples from in-distribution sources, i.e., fastMRI knee, fastMRI brain, CMRxRecon2023 and M4Raw are retained. Filtering affects almost exclusively the 3D MRI data sources which are used as auxiliary training data for improving performance. This observations indicates that an effective filter identifies in-distribution data as much as possible and mainly removes data from auxiliary data sources.

Based on this observation, we now examine how the initial composition of the unfiltered dataset influences the effectiveness of alignment filtering. Figure 7 compares at 8-fold acceleration in-distribution performance between weighted alignment filtering and no filtering as a function of the fraction of in-distribution data in an unfiltered dataset of fixed size (120k slices). We observe that filtering improves performance when the fraction of in-distribution data is low, and in the case where no auxiliary data is used for training, filtering can hurt performance. This suggests that filtering is beneficial when in-distribution data is scarce.

**Dataset size.** Next, we study how filtering performance is related to the size of the unfiltered dataset. Figure 8 compares at 8-fold acceleration in-distribution performance between weighted alignment filtering and no filtering as a function of the unfiltered dataset size containing 10% in-distribution data. We observe that weighted alignment filtering yields similar performance improvements across different data scales. On the investigated data scale, the performance gains obtained by filtering are comparable to the gains obtained by a 3-fold increase of the unfiltered dataset.

**Acceleration factor.** Lastly, we investigate how filtering performance is affected by the acceleration factor, which changes the reconstruction difficulty. Figure 9 shows performance of weighted alignment filtering as a function of the acceleration factor. The unfiltered dataset size is fixed to 120k slices with 1% in-distribution data. We observe that filtering improves performance across acceleration factors with a slight tendency of larger improvements at higher accelerations, where there is more room for improvement.

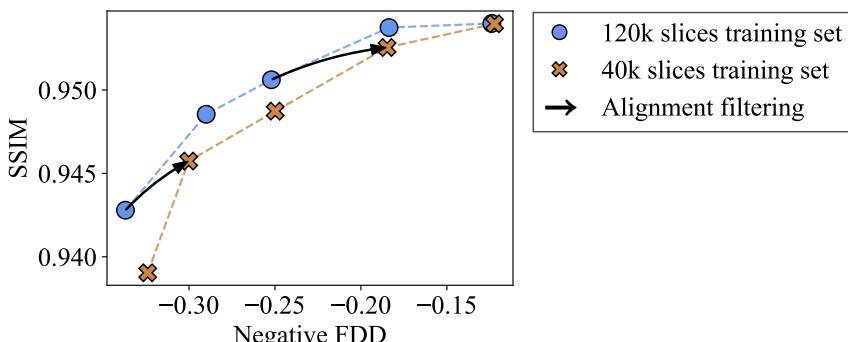

Figure 10: For a fixed training set size, similarity between training and test distribution measured as the negative Fréchet DreamSim Distance (FDD) correlates with performance on the test set. Alignment filtering reduces the dataset size but improves this similarity which relates to improved performance.

Similar qualitative results are obtained when investigating scaling for 4-fold accelerated MRI and scaling with model size. Results are provided in Appendix C.

### 3.5 Relation between performance and train-test set similarity

We hypothesize that alignment filtering, which selects training samples using a validation set that resembles the test set, improves performance by increasing the similarity between the resulting training and the test distribution.

To investigate this hypothesis, we quantify train-test set similarity with what we call the **Fréchet DreamSim Distance**, which is similar to the Fréchet Inception Distance (FID) [16]. Instead of using Inception-v3 embeddings, Fréchet DreamSim Distance uses DreamSim embeddings, following Stein et al. [40], who show that relying on DreamSim embeddings when computing the Fréchet distance between two datasets captures distributional similarity better than relying Inception-v3 embeddings.

Figure 10 shows this metric between training sets and the in-distribution validation sets, and we see a high correlation with reconstruction performance at 4-fold acceleration for fixed training set sizes. Alignment filtering reduces the training set size but increases the train-test similarity which relates to performance gains.

However, while we find that this similarity metric correlates well for in-distribution evaluations, we only observed weak correlation when considering out-of-distribution setups. For example, we found that taking a training set that is completely out-of-distribution relative to the test sets and substituting 1% of that training set with in-distribution data can significantly enhance performance on the in-distribution test sets. Yet, the Fréchet DreamSim Distance remains largely unchanged as only a tiny fraction of the dataset has changed. For out-of-distribution evaluation other similarity metrics, such as those relying on nearest neighbors between training and test sets [26, 22], can provider better correlation with performance.

### 3.6 Results for reconstruction with diffusion models

In the previous sections, we studied data filtering for models trained end-to-end. In this section, we explore whether the same filtering techniques can also benefit diffusion model-based reconstruction approaches for accelerated MRI. We compare diffusion models trained on an unfiltered dataset with those trained on the weighted alignment filtered dataset. The diffusion models are trained on MVUE reconstructions of fully sampled k-space data. For reconstruction, we consider variational optimization [25] and decomposed diffusion sampling [6] (more details are in Appendix D).

Figure 11 shows for 4-fold accelerated MRI that filtering with weighted alignment also improves the performance of diffusion models. This improvement is consistent across both sampling techniques, different sizes of unfiltered data and varying proportions of in-distribution data.

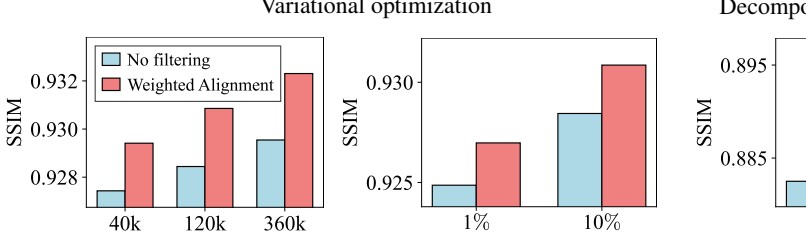

Figure 11: Performance comparison under 4-fold acceleration on in-distribution data of a diffusion model trained on a weighted alignment filtered dataset and a model trained on an unfiltered dataset. We consider variational optimization [25] and decomposed diffusion sampling [6] for reconstruction. We study performance under different sizes of the unfiltered dataset, and varying fraction of in-distribution (ID) data in a fixed size unfiltered dataset of 120k slices. Filtering improves performance of diffusion models with either reconstruction method and across dataset compositions.

## 4  Conclusion and limitations

This work proposes and investigates various filtering strategies for accelerated MRI and demonstrates that data filtering can advance the performance of existing state-of-the-art neural networks.

Our main finding is that data curation through filtering for accelerated 2D MRI consistently improves performance for end-to-end models as well as for diffusion models, which are currently the two most performant and widely used model classes. However, the improvements are relatively modest compared to the improvements data filtering achieves in other domains, e.g., for language models and for vision-language models. The reason could be that the quality of the images in the medical datasets we considered are already of relatively high quality.

In this work we focused on accelerated 2D MRI, while other important related reconstruction problems such as accelerated 3D MRI, motion compensated MRI reconstruction, and image reconstruction problems beyond MRI are not considered.

While refining data curation processes have become critical research areas in machine learning for computer vision and natural language processing, in imaging they received little attention. Our work is an early step towards understanding effective data filtering for imaging, in particular for accelerated MRI.

## Acknowledgments and Disclosure of Funding

The authors acknowledge the financial support by the Federal Ministry of Education and Research of Germany in the programme of "Souverän. Digital. Vernetzt.". Joint project 6G-life, project identification number: 16KISK002, as well as from Deutsche Forschungsgemeinschaft (DFG, German Research Foundation) - 456465471, 464123524, and 517586365.

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

Table 2: Performance measured in PSNR [dB] (↑) and LPIPS (↓) of different filtering methods at 4-fold acceleration, and 120k slices in the unfiltered dataset with 1% in-distribution data.

| Filtering strategy | Dataset size | fastMRI knee | fastMRI brain | In-distribution | Out-of-distribution | Mean over 48 datasets |
|---|---|---|---|---|---|---|
| No filtering | 120k | 40.11 | 40.54 | 39.25 | 40.57 | 39.99 |
| | | 0.155 | 0.103 | 0.115 | 0.108 | 0.111 |
| Heuristic | 80k | 40.25 | 40.75 | 39.50 | 40.74 | 40.19 |
| | | 0.152 | 0.101 | 0.112 | 0.106 | 0.109 |
| Alignment | 40k | 40.38 | 40.96 | 39.74 | 40.88 | 40.38 |
| | | **0.150** | **0.100** | **0.111** | 0.104 | **0.107** |
| Heuristic→Alignment | 40k | 40.40 | 41.00 | 39.77 | 40.94 | 40.42 |
| | | 0.152 | **0.100** | 0.112 | 0.104 | **0.107** |
| Weighted Alignment | 40k | **40.60** | **41.31** | **40.05** | **41.03** | **40.60** |
| | | 0.151 | 0.103 | 0.113 | 0.104 | 0.108 |

# A    Details on the experimental setup

**Code.**    The code for this work can be found here: https://github.com/MLI-lab/data_filtering_for_accelerated_mri. The repository also contains the raw evaluation output data analyzed in this work.

**Access to datasets.**    Due to licensing restrictions from the original dataset sources, we unfortunately cannot host the curated datasets ourselves. However, all datasets used in this work are publicly available from their respective source (see Table 1). We provide code to convert these datasets into a unified format used throughout this work.

**Data conversion.**    Different sources store k-space data in different formats. We organize and save the data with the fastMRI convention, where each k-space volume has shape [number of slices, number of coils, ky, kx] and is stored in a HDF5 file. We split scans that originally included more dimensions, for example, due to multiple echoes (e.g. SKM-TEA [7]) or temporal frames as in cine MRI [44], along those dimensions and treat them as separate volumes. For 3D MRI scans, the k-space data is converted into three distinct volumes, each corresponding to a coronal, axial, or sagittal view. Storing the data from all sources in Table 1 after conversion requires 20TB of disk space.

**Models.**    We rely on the end-to-end VarNet [39] implementation provided by the fastMRI repository. We consider VarNets with 80M parameters that have eight cascades where each reconstruction U-net has 36 channels in the first pooling layer and 4 pooling layers. The original VarNet implementation maps the predicted k-space to a root-sum-of-square reconstruction. However, since we evaluate on MVUE ground-truths, we perform a MVUE reconstruction with the predicted k-space.

**Training.**    We train the VarNets until saturating performance is reached on the validation set. We use the Adam optimizer with $\beta_1 = 0.9, \beta_2 = 0.999$ and a batch size of two. The learning rate is warmed up linearly to 4e-4 using 1% of total training time and then linearly decayed to 1.6e-5. Training a model on an unfiltered dataset of 120k slices using a single NVIDIA L40 GPU and four workers takes around 90 hours and 43GiB in GPU memory. Using the same setup, training a model on an unfiltered dataset of 40k slices takes around 36 hours, on an unfiltered dataset of 360k slices around 170 hours, and on the entire data pool totaling 1.1M slices around 500 hours.

**Evaluation.**    To compute the mean performance score, we compute the average reconstruction performance for each data distribution and then average these scores over all considered data distributions. Moreover, we use the sensitivity maps to compute a mask that better captures the region of interest. This mask is then applied to both the model output and the ground truth to exclude the background before computing a performance metric. This approach reduces variations in the metric caused by reconstruction errors in the background, which are not relevant for evaluation. Moreover, following Lin and Heckel [22], we normalize the reconstructions to have the same mean and variance as the reference image. This reduces metric fluctuations caused by minor, imperceptible differences in brightness and contrast that could otherwise disproportionately impact scores.

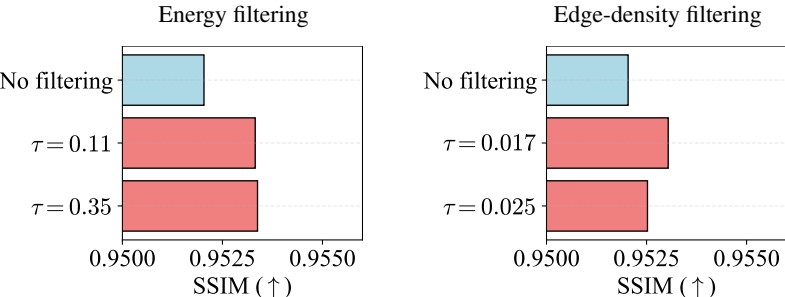

Figure 12: Ablation study for the energy (left) and edge-density (right) thresholds of our heuristic filtering methods. The results show that both heuristic filters provide improvement in SSIM over no filtering. Our chosen thresholds ($\tau = 0.11$ for energy and $\tau = 0.017$ for edge-density) yield the optimal performance.

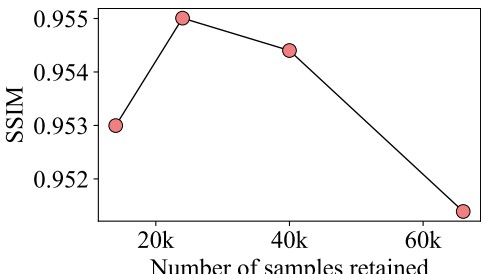

Figure 13: We ablate the choice of the number of nearest neighbors for alignment filtering on a random subset of 120k slices for 4-fold acceleration. Choosing the number of nearest neighbors such that between 20k and 40k samples are retained yields best performance.

**Edge-density filtering.**    We use scikit-image's implementation of the Canny edge detector with the following configuration: `skimage.feature.canny(image, sigma=2, low_threshold=0.01, high_threshold=0.2)`.

**Threshold of heuristic filtering.**    We perform an ablation study on the thresholds for our heuristic filtering methods, as shown in Figure 12. For the energy threshold, we compare the threshold $\tau = 0.11$ against a higher threshold of $\tau = 0.35$ and the no filtering baseline. Both thresholds achieve a similar SSIM and better than baseline; we selected $\tau = 0.11$ because $\tau = 0.35$ resulted in the removal of many slices that contained clear signals, which defeats the purpose of the energy filter. For the edge-density threshold, we test our choice of $\tau = 0.017$ against $\tau = 0.025$ and the no filtering baseline. The chosen value of $\tau = 0.017$ outperforms both the no filtering and the alternative threshold of $\tau = 0.025$. However, these heuristic filtering methods are less effective than the DreamSim-based alignment filtering presented in Section 3.3.

**Number of nearest neighbors for alignment filtering.**    In the main body, we choose the number of nearest neighbor for alignment filtering such that 33% of the data is retained. We ablate this choice for 4-fold acceleration on the unfiltered dataset with 120k slices. Figure 13 shows that choosing the number of nearest neighbors such that between 20k and 40k (33%) samples are retained yields best performance. Based on this observation, we always retain 33% of the data when applying alignment filtering when the unfiltered dataset contains at least 120k slices. For unfiltered datasets with 40k slices the number of nearest neighbors is chosen such that 20k (50%) samples are retained as this choice yielded better results than retaining 33% of the data.

## B    Additional details and results for Section 3.3

**Confidence intervals.**    We use bootstrapping to compute the confidence intervals in Figure 3. We first compute and store the SSIM difference obtained by the model trained on a filtered over the

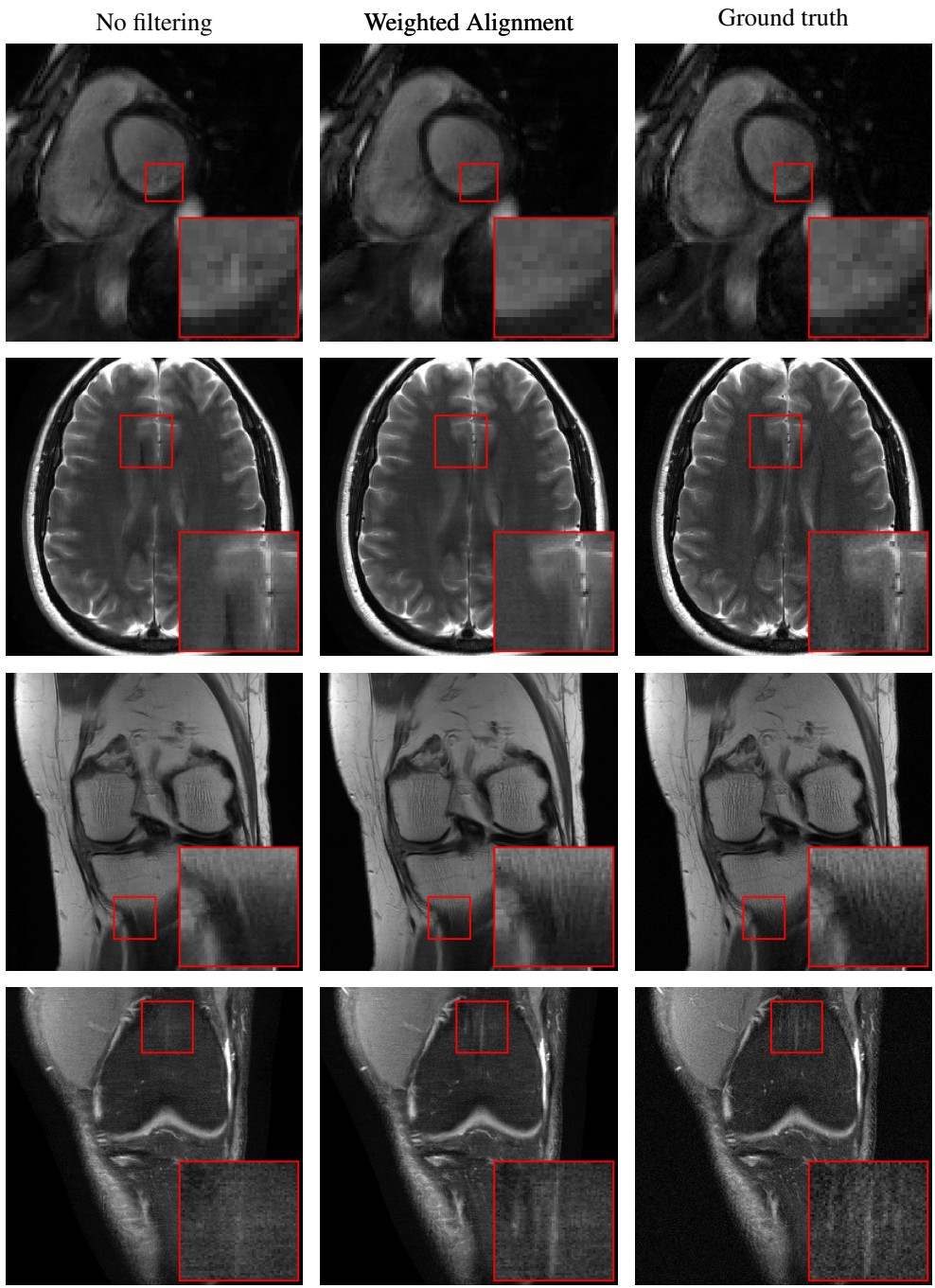

Figure 14: Reconstruction examples at 4-fold acceleration showing reduced artifacts and sharper details in the reconstructions obtained with weighted alignment filtering compared to those obtained with no filtering.

model trained on the unfiltered dataset for each individual reconstruction. From this set of SSIM differences, we sample scores with replacement until we obtain the size of the original test set. Then, we compute the mean SSIM difference by computing first the average SSIM difference for each data distribution and then average these over all considered data distributions. This process is repeated 10000 times which yields a distribution of mean SSIM differences. Finally, from this distribution, we take the 2.5 percentile as lower bound and the 97.5 percentile as upper bound for reporting the 95% confidence interval.

No filtering       Weighted Alignment       Ground truth

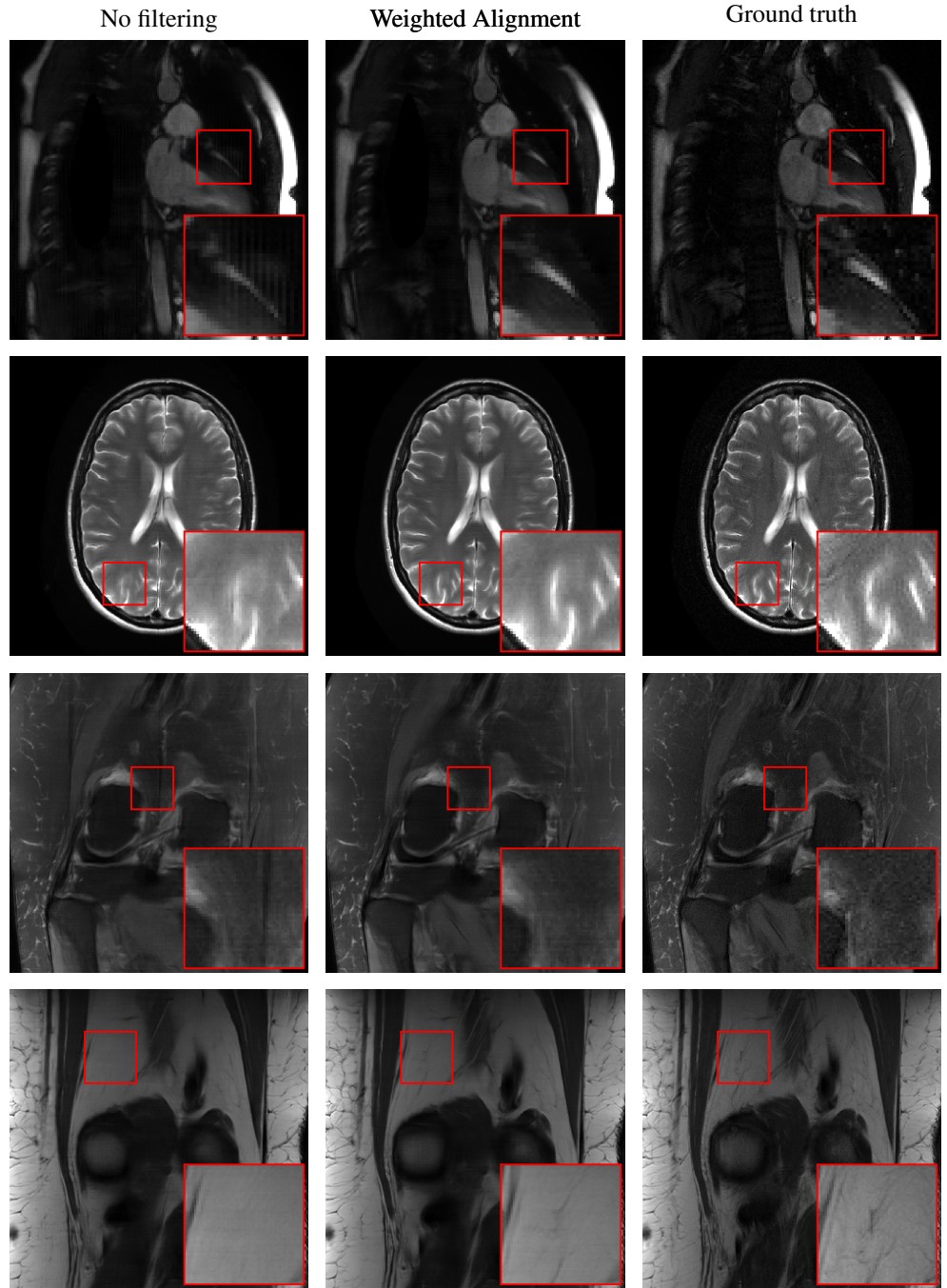

Figure 15: Reconstruction examples at 8-fold acceleration showing reduced artifacts and sharper details in the reconstructions obtained with weighted alignment filtering compared to those obtained with no filtering.

**Additional metrics.**    In Section 3.3, we use SSIM as performance metric for comparing different filtering methods. Table 2 provides additional performance metrics: PSNR and LPIPS [48]. LPIPS is a metric based on features of a pretrained neural network. We include LPIPS because studies have shown that LPIPS correlates well with radiologist readings [1]. Interestingly, we observe a trade-off between weighted alignment filtering and alignment filtering: Weighted alignment filtering obtains higher PSNR but lower LPIPS compared to alignment filtering.

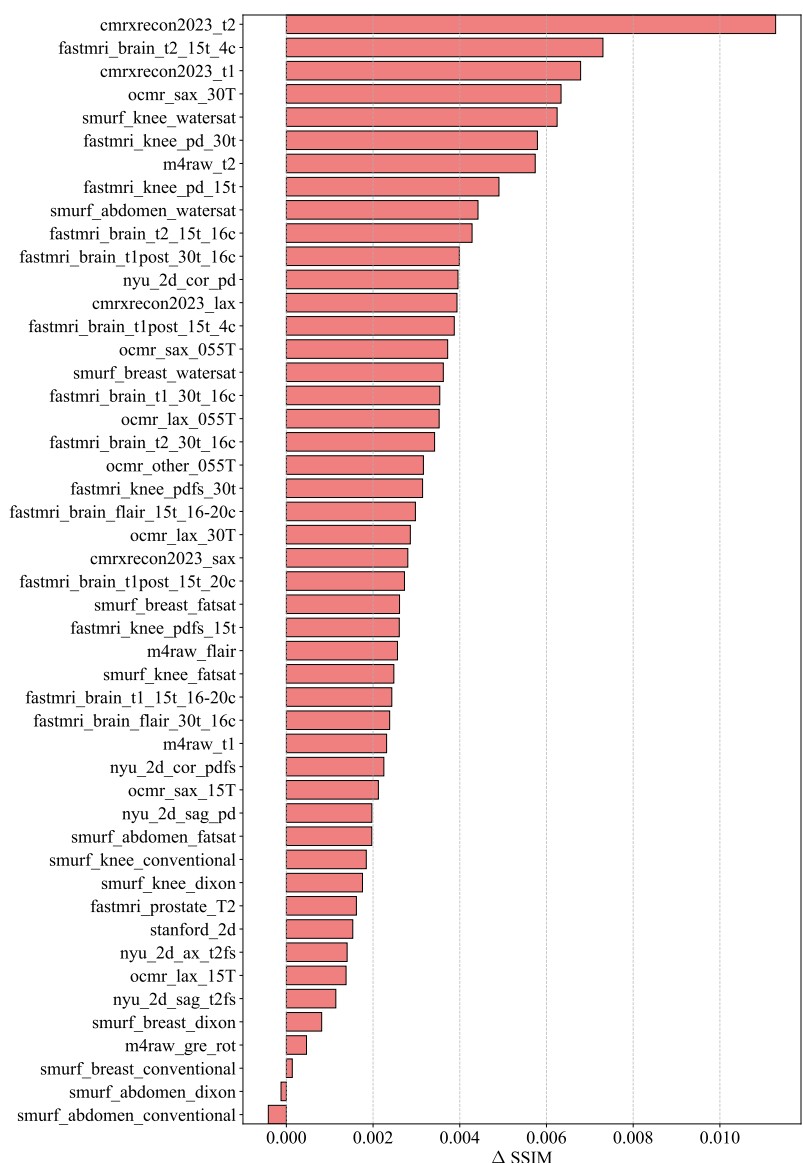

Figure 16: Weighted alignment filtering improves on 46 out of 48 sets for 4-fold accelerated MRI and a dataset size of 120k slices.

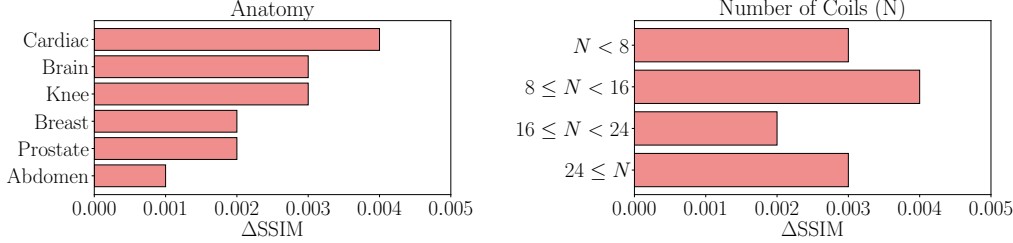

Figure 17: Breakdown of the average SSIM improvement from weighted alignment filtering across subgroups for anatomy, and number of coils. The experimental setup is the same as Figure 16: Weighted alignment filtering on a 120k-slice unfiltered dataset at 4-fold acceleration.

**Additional reconstructions.** Figure 14 (4-fold acceleration) and Figure 15 (8-fold acceleration) provide additional reconstruction examples for the models reported in Section 3.3, further demon-

Table 3: Filtering results for U-net and ViT trained for 4-fold acceleration, and 120k slices in the unfiltered dataset. The unfiltered dataset for training U-net contains 1% in-distribution data and for ViT 10%. Performance is measured in SSIM(↑), PSNR [dB](↑) and LPIPS(↓).

| Model | Filtering strategy | Dataset size | fastMRI knee | fastMRI brain | In-distribution | Out-of-distribution | Mean over 48 datasets |
|---|---|---|---|---|---|---|---|
| U-net 120M param. | No filtering | 120k | 0.905 | 0.935 | 0.909 | **0.911** | 0.910 |
| | | | 36.81 | 36.33 | 35.29 | 35.42 | 35.36 |
| | | | 0.235 | 0.170 | 0.185 | 0.186 | 0.186 |
| | Weighted Alignment | 40k | **0.911** | **0.941** | **0.916** | **0.911** | **0.913** |
| | | | **37.58** | **37.28** | **36.05** | **35.50** | **35.74** |
| | | | **0.220** | **0.162** | **0.177** | **0.184** | **0.181** |
| ViT 60M param. | No filtering | 120k | 0.916 | 0.948 | 0.923 | **0.918** | 0.920 |
| | | | 37.81 | 37.52 | 36.36 | 35.88 | 36.09 |
| | | | 0.212 | 0.158 | 0.170 | 0.178 | 0.174 |
| | Weighted Alignment | 40k | **0.922** | **0.954** | **0.929** | **0.918** | **0.923** |
| | | | **38.41** | **38.54** | **37.15** | **35.97** | **36.49** |
| | | | **0.202** | **0.148** | **0.160** | **0.177** | **0.169** |

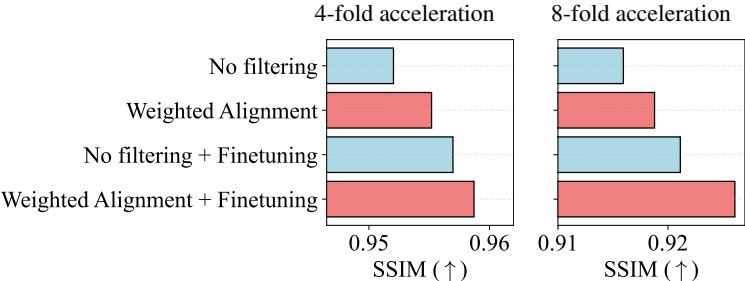

Figure 18: Weighted alignment filtering outperforms no filtering also after the models pretrained on either datasets are fine-tuned on the validation set (269 slices) used for alignment filtering. The unfiltered dataset size is 120k slices.

strating that models trained on the weighted alignment filtered datasets reduce artifacts and produce slightly sharper details compared to those trained on the unfiltered dataset.

**Evaluation on each test set.** In the main body, we report aggregated performance scores. Figure 16 provides for 4-fold accelerated MRI a detailed performance comparison between weighted alignment filtering and no filtering on all 48 test sets. Reported is the difference in SSIM between weighted alignment filtering and no filtering. Filtering yields improvements on 46 out of 48 sets. Dataset names follow the format: <dataset source >_<contrast >_<magnet strength >_<number of coils >.

**Evaluation on different subgroups.** We provide a detailed breakdown of the average SSIM improvement across different subgroups for anatomy, field strength, and number of coils. The results are shown in Figure 17, and it can be seen that all subgroups show improvements. We observe that the most gains were achieved in cardiac scans when evaluating different anatomies. For coil numbers, the highest gains are obtained between 8 and 16 coils, but a global trend cannot be concluded across the entire spectrum of coil numbers considered.

**Other model architectures.** Beside VarNet [39], which is an unrolled network relying on data consistency, we investigate a standard U-net trained for accelerated MRI [47] and a Vision Transformer (ViT) adjusted for accelerated MRI reconstruction [21]. Those two models do not rely on data consistency. While the overall performance is lower than that of VarNet, Table 3 shows that weighted alignment filtering improves performance over no filtering also for those models with gains up to 1dB in PSNR.

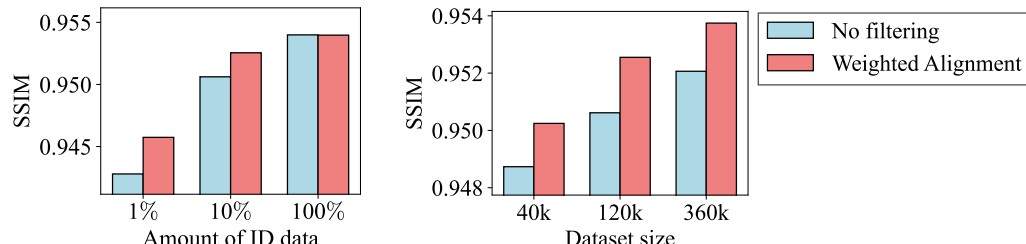

Figure 19: Scaling results (in-distribution performance) at 4-fold acceleration. **Left:** Performance as a function of the amount of in-distribution data in the unfiltered dataset. The unfiltered dataset size is fixed at 120k slices. Filtering improves performance when little in-distribution data is available. **Right:** Performance as a function of the amount total data in the unfiltered dataset. The unfiltered datasets contain 10% in-distribution data. Performance improvements are consistent over different data scales.

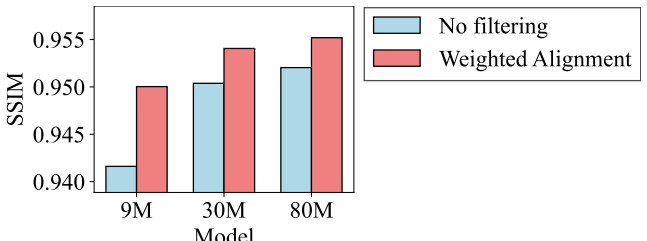

Figure 20: Filtering improves performance across models sizes. The unfiltered dataset contains 120k slices with 1% in-distribution data. The 30M parameter model is fastMRI's default VarNet configuration [39]. Filtering is particularly beneficial for the small VarNet with 9M parameters.

**Fine-tuning.** Alignment filtering relies on a validation set to retrieve images from a data pool that are similar to the evaluation data. We investigate how further fine-tuning on our validation set (269 slices) changes the performance difference between pretraining on the unfiltered dataset and pretraining on the weighted alignment filtered dataset. For each pretrained model, we perform grid search across number of fine-tuning epochs and learning rates and report the best performance obtained on our evaluation set. Figure 18 shows for 4-fold and 8-fold acceleration that weighted alignment filtering outperforms no filtering also after the models pretrained on either datasets are fine-tuned on the validation set.

## C  Additional details and results for Section 3.4

**Scaling experiments.** In Section 3.4, we report results on scaling-experiments for 8-fold acceleration where we investigate in-distribution performance as a function of in-distribution data proportion (Figure 7) and as a function of dataset size (Figure 8). Figure 19 reports the result on the same scaling-experiments but for 4-fold acceleration and using alignment filtering. Also here, filtering improves performance across dataset sizes and for low in-distribution data proportions.

**Model size.** Figure 20 shows that weighted alignment filtering improves performance across models sizes. The unfiltered dataset contains 120k slices with 1% in-distribution data. Filtering is particularly beneficial for the small VarNet with 9M parameters.

**Retrieval Metric.** Table 4 shows that DreamSim outperforms a pixel-based metric for filtering. The comparison is based on an unfiltered dataset of 40k slices for 8-fold accelerated MRI, where the pixel-level metric is the Euclidean distance. The pixel-based approach is ineffective on the cardiac test sets, resulting in performance degradation.

**t-SNE visualization.** To provide a visual interpretation of the filtering process, we applied t-SNE [24] to the DreamSim embeddings of the 120k slices dataset before and after alignment filtering,

Table 4: Comparison of DreamSim and a pixel-based metric (Euclidean distance) for alignment filtering on 40k slices with 8-fold acceleration. Numbers indicate the SSIM gain over no filtering. DreamSim shows consistent positive improvement in SSIM over no filtering. In contrast, the pixel-based approach performs poorly on the cardiac subset and yields no average improvement across all test sets.

| Filtering strategy | Cardiac | non-Cardiac | All test sets |
|---|---|---|---|
| Pixel-based | −0.010 | +0.003 | 0.0 |
| DreamSim | +0.006 | +0.006 | +0.006 |

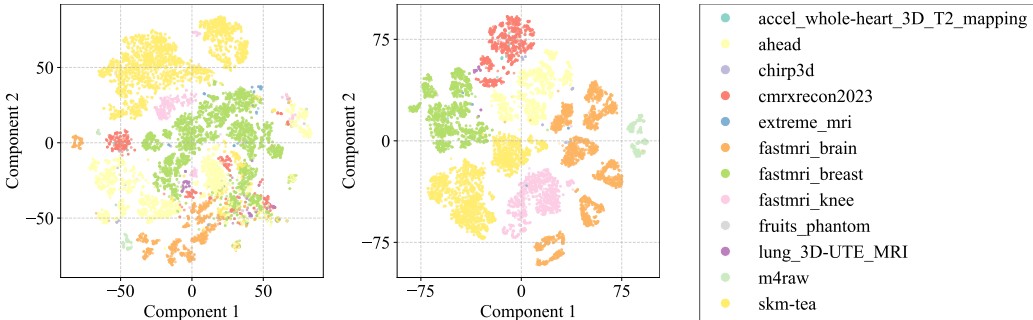

Figure 21: t-SNE visualization of DreamSim embeddings on the 120k-slice dataset, before (left) and after (right) applying alignment filtering. Each color corresponds to a different data source from the initial unfiltered data pool. The filtered dataset exhibits more distinct and separated clusters.

as shown in Figure 21. The unfiltered dataset (left) shows that embeddings from different data sources exhibit significant overlap. In contrast, the filtered dataset (right) displays considerably more distinct and well-separated clusters.

# D  Additional details for Section 3.6

In Section 3.6, we extend our results to diffusion model-based MRI reconstruction. This section provides background and implementation details for the diffusion models that we consider.

**Background.** A diffusion model $\epsilon_\theta$ with parameters $\theta$ aims to learn a data distribution $p(\mathbf{x})$. Diffusion models consist of a forward process which gradually adds noise to images from the distribution $p(\mathbf{x})$, and a reverse process which aims to invert forward process. We adopt the denoising diffusion probabilistic models (DDPM) formulation [17], where each step of the forward process is Gaussian distributed $p(\mathbf{x}_{t+1}|\mathbf{x}_t) = \mathcal{N}(\sqrt{1-\beta_t}\mathbf{x}_t, \beta_t^2 \mathbf{I})$ for time steps $t = 0, 1, \ldots, 1000$ and increasing noise levels $\beta_t$. The reverse diffusion process is modeled with Gaussian transition probabilities $p(\mathbf{x}_{t-1}|\mathbf{x}_t)$, and the mean of the Gaussian is learned with a neural network. The diffusion model $\epsilon_\theta(\mathbf{x}; t)$ is trained using the residual denoising objective

$$\mathcal{L}(\theta, \mathbf{x}) = \mathbb{E}_{t\sim\mathcal{U}(0,T), \epsilon\sim\mathcal{N}(0,\mathbf{I})} \left[ \left\| \epsilon_\theta(\sqrt{1-\sigma_t^2}\mathbf{x} + \sigma_t\epsilon; t) - \epsilon \right\|_2^2 \right],$$

where $\sigma_t^2 = 1 - \Pi_{s=1}^t (1 - \beta_s)$. Samples from the distribution $p(\mathbf{x}_0)$ can then be obtained by sampling $\mathbf{x}_T \sim \mathcal{N}(0, \mathbf{I})$ and successively applying the learned Gaussian transitions $p(\mathbf{x}_{t-1}|\mathbf{x}_t)$. Unlike end-to-end models, diffusion models do not require measurement data for training, but enforce data-consistency during reconstruction using the diffusion model as a pretrained image prior.

In our setup, the diffusion models learn the data distribution of the fully-sampled MVUE reconstructions. For the diffusion model, we choose the U-Net [34] architecture adopted from [8] with 80M parameters.

Table 5: Datasets used for accelerated 3D MRI setup. First two correspond to in-distribution datasets and the last two are used for out-of-distribution evaluation.

| Dataset | Anatomy | View | Image contrast | Vendor | Magnet | Coils | Vol./Subj. |
|---|---|---|---|---|---|---|---|
| SKM-TEA [7] | knee | various | qDESS | GE | 3T | 8, 16 | 930/155 |
| AHEAD [3] | brain | various | MP2RAGE-ME | Philips | 7T | 32 | 1.1k/77 |
| Stanford-3D [9] | knee | various | CUBE (3D-FSE) | GE | 3T | 16 | 19/19 |
| CC359 [38] | brain | various | GRE | GE | 3T | 32 | 165/165 |

Accelerated MRI is modeled as a linear inverse problem of the form $\mathbf{y} = \mathbf{A}\mathbf{x} + \mathbf{z}$, with linear forward operator $\mathbf{A}$ and additive Gaussian noise $\mathbf{z}$. We consider the following two approaches for reconstruction with diffusion models.

**Posterior sampling.** We consider decomposed diffusion sampling [6]. Assuming that $\mathbf{x}$ is drawn from the true data distribution of MVUE reconstructions, solving the inverse problem consists of sampling from the posterior $p(\mathbf{x}_0|\mathbf{y})$. Diffusion models enable posterior sampling by conditioning the reverse process $p(\mathbf{x}_{t-1}|\mathbf{x}_t, \hat{\mathbf{x}}_0(\mathbf{x}_t), \mathbf{y})$ on the measurements $\mathbf{y}$. In each step of the reverse sampling process, decomposed diffusion sampling updates the denoised estimate $\hat{\mathbf{x}}_0(\mathbf{x}_t)$ by minimizing $\frac{\gamma}{2}\|\mathbf{y} - \mathbf{A}\mathbf{x}\|_2^2 + \frac{1}{2}\|\mathbf{x} - \hat{\mathbf{x}}_0(\mathbf{x}_t)\|_2^2$. This allows to control the influence of the diffusion prior via the estimate $\hat{\mathbf{x}}_0(\mathbf{x}_t)$. Moreover, we use denoising diffusion implicit model sampling to accelerate the sampling process [37].

**Variational approach.** We consider the approach proposed by Mardani et al. [25], which consists of solving $\min_q KL(q(\mathbf{x}_0|y), p(\mathbf{x}_0|\mathbf{y}))$, with a variational distribution $q = \mathcal{N}(\mu, \sigma^2)$. This motivates the following variational objective:

$$\hat{\mathbf{x}}(\mathbf{y}) = \underset{\mathbf{x}\in\mathbb{C}^N}{\mathrm{argmin}}\|\mathbf{y} - \mathbf{A}\mathbf{x}\|_2^2 + \lambda\mathbb{E}_{t\sim\mathcal{U}(0,T'),\boldsymbol{\epsilon}\sim\mathcal{N}(0,\mathbf{I})}\left[w(t)\left\|\boldsymbol{\epsilon}_\theta(\sqrt{1-\sigma_t^2}\mathbf{x} + \sigma_t\boldsymbol{\epsilon}; t) - \boldsymbol{\epsilon}\right\|_2^2\right],$$

where $\lambda$ is a hyperparameter. We choose the time step dependent weighting factor $w(t)$ following [25]. The measurements $\mathbf{y}$ are scaled such that the reconstruction $\mathbf{x}$ has approximately unit variance. We minimize the objective using a first order gradient optimizer, and initialize with the zero-filled least-square reconstruction. Finally, we perform uniform time step sampling with upper bound $T' = 0.4 \cdot T$ (similar to [18]).

**Choice of hyperparameters.** The performance of diffusion models for reconstruction is strongly dependent on hyperparameter choices. For example, the performance of the variational approach critically depends on the choice of the regularization parameter $\lambda$. To more confidently attribute performance differences to variations in dataset design rather than suboptimal hyperparameter choices, we study diffusion models under best-case conditions: For both the posterior sampling and the variational approach for reconstruction with diffusion models, we tune hyperparameters for each sample in the test set individually with a grid search based on the ground-truth image.

**Training.** Similar to the end-to-end models we train the diffusion models until saturating reconstruction performance is reached on the validation set. We use the Adam optimizer with $\beta_1 = 0.9, \beta_2 = 0.999$, and a batch size of two. The learning rate is warmed up linearly to 4e-4 using 1% of total training time and then linearly decayed to 1.6e-5. Compute resources when using four workers are similar to the end-to-end experiments (Appendix A).

### D.1 Results for 3D MRI reconstruction

In the following, we investigate filtering for diffusion model-based 3D MRI reconstruction. In previous sections, we used 3D MRI datasets only as auxiliary data sources for improving 2D MRI reconstruction performance. However, in this subsection we perform 3D reconstruction with 3D undersampling masks on a curated set of 3D volumes.

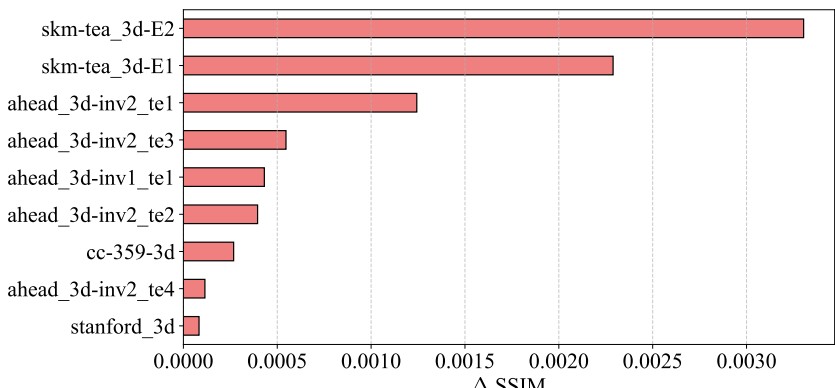

Figure 22: Detailed evaluation of weighted alignment filtering for 3D reconstruction performance at $36\times$ acceleration. Similar to the 2D reconstruction results presented in Fig 16, we find that gains via filtering are larger on in-distribution samples (AHEAD and SKM-TEA) than on out-of-distribution samples (Stanford 3D and CC-359).

**Variational 3D MRI reconstruction.** We perform 3D reconstruction using 2D diffusion models trained on complex-valued MVUE reconstructions, and using a variational approach, where the diffusion model is applied to regularize randomly selected slices [18]. We minimize the following objective using gradient descent:

$$\hat{\mathbf{x}}(\mathbf{y}) = \underset{\mathbf{x}\in\mathbb{C}^N}{\operatorname{argmin}} \sum_{i=1}^{C} \|\mathbf{y}_i - \mathbf{MF}_{3D}\mathbf{S}_i\mathbf{x}\|_2^2$$
$$+ \lambda \mathbb{E}_{\mathbf{s}\sim\text{2D-Slices}(\mathbf{x})} \left[ \mathbb{E}_{t\sim\mathcal{U}(0,T'),\boldsymbol{\epsilon}\sim\mathcal{N}(0,\mathbf{I})} \left[ w(t) \left\| \boldsymbol{\epsilon}_\theta(\sqrt{1-\sigma_t^2}\mathbf{s} + \sigma_t\boldsymbol{\epsilon}; t) - \boldsymbol{\epsilon} \right\|_2^2 \right] \right].$$

Here, $\mathbf{S}_i$ encodes the sensitivity map associated with the $i$-th receiver coil, $\mathbf{F}_{3D}$ is the 3D discrete Fourier transform, and $\mathbf{M}$ a 2D hybrid-cartesian Poisson undersampling mask in our experiments. Moreover, we employ a pre-trained 2D complex diffusion model $\boldsymbol{\epsilon}_\phi(\mathbf{x}_t, t)$. We follow the same instance-specific hyperparameter tuning method for $\lambda$ as for 2D reconstruction, and approximate the expectation with respect to random slices by uniformly sampling 50 slices per anatomical view and gradient descent iteration.

We follow the same training setup as for 2D diffusion models, but scale the slices by the norm of the 3D volume during training.

**Evaluation set.** To evaluate the reconstruction performance on 3D MRI we curate a diverse set of 3D MRI volumes based on the datasets stated in Table 5. SKM-TEA and AHEAD contain two and five echoes, respectively. We split SKM-TEA into two subsets, one for each echo, and perform a similar split with the AHEAD dataset. During curation, we excluded many volumes with artifacts, such as wraparound or de-identification artifacts apparent in brain datasets. In total, our validation dataset consists of 30 in-distribution volumes and 9 out-of-distribution volumes.

**Filtering.** We perform weighted alignment filtering using a validation set similarly curated as the 3D evaluation dataset. We adapt the filtering method to 3D, by randomly selecting slices along all anatomical planes of the volumes, while excluding slices near the boundaries.

**Results on datasets from the main body.** We train a diffusion model on the unfiltered 120k slices dataset (same unfiltered dataset as in Section 3.3 for 4-fold accelerated 2D MRI), and train a diffusion model on the filtered dataset with 40k slices retained. Figure 22 provides a detailed evaluation for $36\times$-accelerated MRI. Similar to 2D MRI reconstruction, we observe that the benefit of weighted alignment filtering is larger on in-distribution data than on out-of-distribution datasets. However, on average, our filtering setup benefits 3D reconstruction performance only marginally (+0.001 SSIM) as shown in Figure 23 for 24-fold and 36-fold acceleration; therefore, we cannot conclude that filtering yields meaningful improvements.

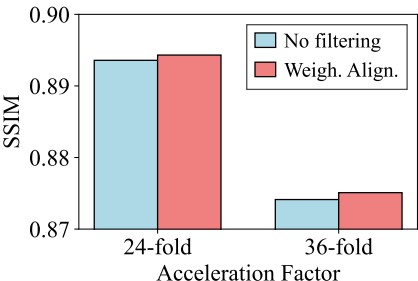
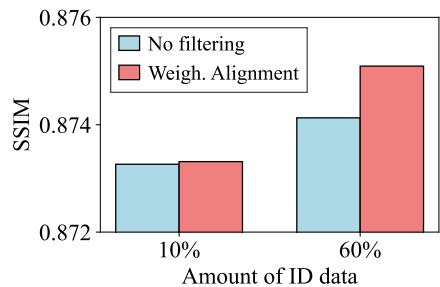

Figure 23: We train one 2D diffusion model on the 120k slices unfiltered dataset (same dataset as in Sec 3.3), and one model on the weighted alignment filtered dataset. We evaluate on a curated set of 3D volumes (see Table 5) for 24-fold acceleration and 36-fold acceleration. The gain obtained by weighted alignment filtering is 0.001 SSIM.

Figure 24: We perform filtering experiments with different fractions of in-distribution data (with respect to our 3D evaluation Table 5) in the unfiltered dataset with 120k slices in total. Different to the 2D MRI experiments, we do not observe improved performance of filtering when the fraction of in-distribution data is low.

**Results for lower amounts of ID data.** Note that the majority of the slices contained in the data pool (see Table 1) used in our work originates from 3D MRI and therefore the unfiltered dataset used in the previous paragraph contains around 60% in-distribution data (with respect to our 3D evaluation). However, in Section 3.4, we observed that filtering can provide a larger benefit when the fraction of in-distribution data in the unfiltered dataset is low. We investigate whether this holds true for our 3D MRI setup and create an unfiltered dataset with reduced in-distribution data (10%) by randomly sampling 10% of the data from 3D volumes with the reaming 90% from 2D volumes, totaling 120k slices. We perform weighted alignment filtering on this created unfiltered dataset, and present the reconstruction results with the correspondingly trained diffusion models in Figure 24. However, different to our results for 2D MRI, we do not find that filtering is more beneficial when the fraction of in-distribution data decreases.

# E   Licenses for datasets and software

- fastMRI datasets [47, 43, 36]: Custom agreement: https://fastmri.med.nyu.edu/
- CMRxRecon2023 [44]: CC-BY
- M4Raw [23, 23]: CC-BY 4.0
- SKM-TEA [7]: Custom agreement: https://stanfordaimi.azurewebsites.net/datasets/4aaeafb9-c6e6-4e3c-9188-3aaaf0e0a9e7
- AHEAD [3]: CC BY 4.0
- Lung 3D UTE [27]: CC0-1.0
- Chirp 3D [31]: CC-BY-4.0
- Extreme MRI [30]: CC-BY-4.0
- Fruits, Phantom [46]: CC-BY-4.0
- Heart T2-mapping [49]: CC0-1.0
- Stanford 2D [5]: CC BY-NC 4.0
- NYU data [15]: CC BY-NC 4.0
- SMURF [2]: CC0-1.0
- OCMR [4]: Custom agreement: https://www.ocmr.info/download/
- fastMRI code [47]: MIT License
- BART toolbox [42]: BSD-3-Clause license

