# OpenReview forum: "Improving Deep Learning for Accelerated MRI With Data Filtering"
_NeurIPS.cc/2025/Datasets_and_Benchmarks_Track — NeurIPS 2025 Datasets and Benchmarks Track poster_

### Official Review · Reviewer_j27b · 2025-06-03

**Rating:** 4
**Confidence:** 3

**Summary:**

This paper investigates how data filtering strategies can improve the performance of deep learning models for accelerated Magnetic Resonance Imaging (MRI) reconstruction. The authors propose a novel data curation approach by combining raw k-space data from 18 publicly available sources and constructing a diverse evaluation suite. They then explore various data filtering methods based on heuristic rules and image similarity using the DreamSim metric. Experimental results show that when in-distribution samples are scarce, data filtering strategies can significantly enhance the reconstruction performance of state-of-the-art neural networks (such as VarNet) and diffusion models across multiple evaluation settings.

**Dataset Code Accessibility:**

Yes

**Dataset Code Comments:**

The GitHub repository provides access to the datasets as well as the complete training and filtering code.

**Ethical Considerations:**

No, there are no or only very minor ethics concerns

**Limitations Weaknesses:**

1.The DreamSim-based filtering strategy relies on a validation set that is similar to the test set, which may be difficult to construct in real-world clinical settings. It is recommended to further explore automatic filtering mechanisms that operate without a validation set or under self-supervised paradigms, in order to enhance the robustness and scalability of the method in practical applications.

2.The filtering process is computationally expensive, as it requires extracting DreamSim embeddings for all training and validation images.

3.While the paper explicitly notes the presence of low-quality images in the test data, the validation set used for guiding the DreamSim-based filtering consists of manually selected high-quality images. As a result, the model's ability to discriminate under MRI-specific artifacts has not been verified. It is recommended to explicitly evaluate the behavior of DreamSim on MRI images containing noise and artifacts.

**Strengths Contributions:**

1.Compared to existing literature that mostly focuses on optimizing model architectures, the authors propose a systematic approach to improving MRI reconstruction performance from the perspective of data.

2.This paper constructs a large-scale evaluation suite comprising 48 test sets, covering both in-distribution and out-of-distribution data, providing strong validation for generalization.

3.The work introduces DreamSim, a perceptual image similarity metric, for filtering training data.

4.Notably, even with a smaller training set (e.g., 40k slices), the filtered dataset outperforms much larger unfiltered datasets (e.g., 120k or 360k), making the method well-suited for real-world medical scenarios where data is limited.

---

> ### Author Rebuttal · Authors · 2025-07-31
>
> Many thanks for reviewing our work. We are glad that the reviewer acknowledges the novelty of our work, our extensive evaluation and the effectiveness of our method.
>
> In the following, we address the concerns raised by the reviewer.
>
> 1. Regarding the concern that our proposed retrieval filtering relies on a “validation set that is similar to the test set, which may be difficult to construct in real-world clinical settings” and the suggestion to explore filtering strategies that don’t require a validation set:
>
> We consider a setup where we have access to training and test data with fully-sampled ground-truth data, similar to [1,2,3]. In such a setup, a validation set, reflecting the test distribution, is typically also acquired for hyperparameter tuning, and in our case used for data filtering. Hence, in a setup like ours, our proposed retrieval filtering uses only available data.
> \
> Nevertheless, we agree that filtering strategies that do not leverage a validation set are appealing, especially in setups where access to ground-truth images is limited (such as motion compensated MRI as mentioned in our limitations section), and for this, we have explored heuristic filters described in Section 3.1. Those heuristic filters also yield gains, but less so than filtering with the validation set. However, even such filtering strategies have hyperparameters that need to be tuned, ideally on a validation set.
>
> 2. Regarding that our proposed retrieval filtering is “computationally expensive’’:
>
> The costs of computing the DreamSim embeddings are relatively **minor**, since they only require one forward pass of the dataset through the rather small DreamSim model, which is negligible relative to training.
>
> 3. Regarding the reviewer’s statement that the validation set used for filtering only contains high-quality images while the test set has low-quality images, and the concern that DreamSim's behaviour on MRI images with noise or artifacts is unexplored.
>
> Low-quality images have been removed in **both** test and validation sets (see line 151, where we state that the validation set is constructed in the same fashion as the test set).
> Regarding DreamSim’s performance under noisy images, we find that the DreamSim model retrieves noisy images well: The M4Raw dataset [3] comprises low-field MRI scans which are inherently noisy, yet, as seen in Figure 6, M4Raw images are retrieved very well.
>
> \
> We hope our response has addressed the reviewer’s concerns and in light of the positive feedback, we would be pleased if the reviewer considers raising their score.
>
> \
> **References:**
>
> [1] J. Zbontar et al. “fastMRI: An Open Dataset and Benchmarks for Accelerated MRI.” arXiv:1811.08839, 2019.
>
> [2] C. Wang et al. “CMRxRecon: A publicly available 403 k-space dataset and benchmark to advance deep learning for cardiac MRI.” Scientific Data, 2024.
>
> [3] M. Lyu et al. “M4Raw: A multi-contrast, multi-repetition, multi-channel MRI k-space dataset for low-field.” Scientific Data, 2023.

---

### Official Review · Reviewer_vdoL · 2025-06-27

**Rating:** 5
**Confidence:** 4

**Summary:**

In this work, authors investigate data curation strategies for improved MRI reconstruction. In particular, authors gather a large dataset of raw k-space data from $18$ public datasets and construct a diverse evaluation set comprising of $48$ test sets which capture various anatomies, contrasts, number of coils, etc. Authors then explore various data filtering strategies to enhance the sample efficiency and the performance of current SOTA neural networks. Authors demonstrate that their "weighted retrieval" filtering strategy provides a modest but consistent boost to the performance (for both VarNets and Diffusion Models) while reducing the number of training samples $\approx10$ fold.

**Additional Feedback:**

***Questions and Suggestions:***
* In terms of dataset usage: I think the different dataset yaml files need to be annotated better.  For example, it is not obvious to me what’s the difference between “data_pool_120k.json” vs “data_pool_aux_120k.json”
* In Figure $1$, is the reconstructed image for “no filtering” corresponds to dataset with 0.3% or the 3% ratio on the right?
* In Figure $3$ and $5$, I believe the final size of the Weighted Retrieval based filtered dataset is 40k (Figure 1 caption). It could be useful to mention the number again to emphasize how much reduction there is in the dataset while matching or exceeding the performance of the bigger unfiltered dataset of size $360k$.
* How are the heuristic filtering thresholds of $0.11$ (line $131$) or $0.017$ (line $135$) chosen?
* What’s the performance when the dataset is randomly sampled to match the size of “Weighted Retrieval” filtered dataset? This will highlight the performance gain of smart filtering even better.
* Line $478$-$479$: “higher PSNR, but lower LPIPS” Do the authors have an intuition why that might be the case?
* Figure $3$: why is out-of-distribution performance better than in-distribution performance?

***
***Typos:***
* line $108$: MVUE is not defined in the paper.
* line $109$: bart → BART
* line $444$: GiB or GB?

**Dataset Code Accessibility:**

Yes

**Dataset Code Comments:**

The authors provide a GitHub link: https://github.com/MLI-lab/data_filtering_for_accelerated_mri which has detailed instructions on how to download the dataset and reproduce the numbers in the paper. I have not personally run the code.

**Ethical Comments:**

The authors combine public datasets and provide a filtering pipeline. Since there are no ethical concerns with the public data out there, I don't think authors' filtering pipeline raise additional concerns.

**Ethical Considerations:**

No, there are no or only very minor ethics concerns

**Final Justification:**

Authors provided a new performance break down in terms of anatomy, field strength and number of coils which I think is a good addition to the paper. In general, authors have addressed my concerns/questions successfully. Considering the responses and proposed changes, I revise my rating to $5$.

Note: the computational complexity of obtaining DreamSim embeddings was something that I did not consider initially. I find the discussion between the AC and the authors interesting. Overall, I agree with the authors that 0.6 hours to obtain embeddings of 120k images on a single GPU is not a significant cost.

**Limitations Weaknesses:**

* I don't think the paper has any major weaknesses. However, I feel like the analysis could be more detailed. In particular, authors collect $48$ evaluation sets but eventually the analysis in the main paper is on "in-distribution" and "out-of-distribution" sets. What about the performance on certain anatomies, number-of-coils, field strength? While Figure $12$ provides a breakdown based on $48$ different sets, there is no discussion about it. I understand the authors focus more on the filtering aspect (as evident by the title of the paper), but as part of the dataset track, I believe the readers of this paper would appreciate a more fine-grained analysis.
* The dataset breakdown in Table $1$ indicates that it is much bigger than $360k$ slices used in the experiments. Could you add discussions/motivation on what is the expected performance if all of the data is used? why were the numbers $120k$ and $360k$ picked in particular?
* Is the DreamSim model trained on medical images? Can we be sure that the metrics it spit out for MRI slices are meaningful? I’m particularly interested in the case whether DreamSim can provide meaningful comparison of MVUEs of $8$-fold accelerated measurements. Depending on the answer to these questions, it is possible that the weighted retrieval performance can be improved further with a model that understands medical images better.
* Please see the questions below.

**Strengths Contributions:**

* The paper is written very well. It is organized, easy to understand and follow.
* The curated dataset comprises of many different parameters.
* Besides numerical improvements, many visual evidence of improvements in small details are provided, proving the effectiveness of filtering better.

---

> ### Author Rebuttal · Authors · 2025-07-31
>
> Thank you very much for the detailed review, the helpful suggestions, and for emphasizing the effectiveness of our proposed method.
>
> In the following, we address the concerns raised by the reviewer:
>
> 1. Regarding a more detailed analysis of the evaluation scores such as stating the performance on certain anatomies, number-of-coils, field strength:
>
> Following this suggestion, we analyzed the evaluation scores stated in Figure 12 in more detail as shown in the tables below:
>
> Anatomy:
>
> |               | Cardiac | Brain  | Knee   | Breast | Prostate | Abdomen |
> |:-------------:|:-------:|:------:|:------:|:------:|:--------:|:-------:|
> | SSIM gain| +0.004  | +0.003 | +0.003 | +0.002 | +0.002   | +0.001  |
>
> Field strength:
>
> |               | less than 1.5T | 1.5T   | 3.0T   |
> |:-------------:|:--------------:|:------:|:------:|
> | SSIM gain | +0.003         | +0.003 | +0.003 |
>
> Number of coils (N):
>
> |               | N < 8  | 8 ≤ N < 16 | 16 ≤ N < 24 | 24 ≤ N |
> |:-------------:|:------:|:----------:|:-----------:|:------:|
> | SSIM gain | +0.003 | +0.004     | +0.002      | +0.003 |
>
> We observe that the most gains were achieved in cardiac scans when evaluating different anatomies. Regarding field strengths, we observe uniform improvements. For coil numbers, the highest gains are obtained between 8 and 16 coils, but a global trend cannot be concluded across the entire spectrum of coil numbers considered.
> We will include this analysis in the revised version of the paper.
>
> 2. Regarding filtering experiments when using all the collected data and why we studied subsets:
>
> We have applied our proposed weighted retrieval filtering on the **entire** available data pool (1.1M slices) and found that filtering also **improves** performance at this scale, with similar gains as obtained at smaller scales. This result extends Figure 8 and we will include it in our revised paper. We studied smaller subsets to observe trends for filtering across different scales (Figure 8) and for faster experimentation.
>
> 3. Regarding whether the DreamSim model is trained on medical images and whether this metric is meaningful for MRI.
>
> We use pre-trained DreamSim checkpoints from the original paper [1]. While DreamSim wasn't specifically trained on medical images, Figure 6 shows its effectiveness in retrieving relevant images, correctly identifying most in-distribution samples. This suggests DreamSim is meaningful for retrieval. Our results in Section 3.3 further support this, as we improve performance by filtering data using DreamSim embeddings. Training a new DreamSim model on new medical imaging data would require extensive manual data work [1].
> \
> Regarding “whether DreamSim can provide meaningful comparison of MVUEs of 8-fold accelerated measurements”. To clarify, DreamSim is applied to the fully-sampled (target) MVUE reconstructions, making it agnostic to the acceleration factor. We'll add this clarification to the revised paper.
>
> We now address the questions and suggestions raised by the reviewer:
>
> 1. Regarding the suggestion that the different dataset yaml files should be annotated better in the code base:
>
> We thank the reviewer for this feedback. We will improve the naming convention (e.g., data_pool_[total number of slices]_[fraction of in-indistribution data].json) and add further descriptions, such as the filtering method used, in the .yaml and .json files. Since we are not allowed to make changes during the rebuttal period, we will do it after.
>
> 2. “In Figure 1, does the reconstructed image for ‘no filtering’ correspond to the dataset with 0.3% or the 3% ratio on the right?”
>
> The image is from the CMRxRecon2023 dataset [2], not from the fastMRI knee dataset [3] (as mentioned in the figure caption). The fraction of the CMRxRecon2023 data in the unfiltered dataset is 4%. We use the fastMRI dataset for the plot on the right because it is the most prominent MRI dataset.
>
> 3. “How are the heuristic filtering thresholds chosen?”
>
> For heuristic filtering we provide the following ablations:
> \
> Regarding energy threshold (Th) hyperparameter:
>
> |            | No filtering | Th = 0.11 (our choice) | Th = 0.35 |
> |----------|:--------------:|:-----------------------------:|:--------------:|
> |SSIM   | 0.952         | 0.953                           | 0.953        |
>
> We chose a threshold of 0.11 over 0.35, even though both yield the same performance. The 0.35 threshold resulted in the removal of many slices that contained clear signals, which defeated the purpose of the energy filter.
>
> Regarding edge-density threshold (Th) hyperparameter:
>
> |              | No filtering | Th = 0.017 (our choice) | Th = 0.025 |
> |----------|:--------------:|:-------------------------------:|:-------:|
> |SSIM   | 0.952         | 0.953                              | 0.952         |
>
> However, we note that while heuristic filtering methods were explored, we found the  DreamSim-based retrieval filtering to be more effective (see Section 3.3).
>
> 4. “What’s the performance when the dataset is randomly sampled to match the size of the weighted retrieval filtered dataset? This will highlight the performance gain of filtering even better.”
>
> Figure 8 demonstrates that weighted retrieval filtering applied on the 120k unfiltered dataset, retaining 40k slices, achieves an SSIM of 0.933. In contrast, training on a random subset of 40k slices yields a much lower SSIM of 0.920. To facilitate this comparison, we will update the caption of Figure 8 with: "Weighted retrieval filtering significantly outperforms a randomly selected subset of the same size, as demonstrated by comparing its performance against a same-sized unfiltered dataset."
>
> 5. “Do the authors have an intuition why weighted retrieval filtering provides better PSNR but lower LPIPS compared to retrieval filtering (referring to Table 2)?”
>
> We found that data distributions with inherently noisier data, like those from low-field MRI scans [4], are mostly affected by this observation. Our intuition is as follows: Our training objective is SSIM, which is a pixel-domain similarity metric, like PSNR. Training to increase SSIM or PSNR beyond a certain point could lead to slightly smoother reconstructions as noisy details are hard to reconstruct. In contrast LPIPS, which is a perceptual metric, better aligns with human perception, favoring crisper images despite less pixel-level accuracy. However, we note that other explanations could be valid too.
>
> 6. “Figure 3: why is out-of-distribution performance better than in-distribution performance?”
>
> Performance varies based on the intrinsic reconstruction difficulty. In Figure 3, out-of-distribution test sets are, on average, easier than in-distribution test sets, leading to higher scores. For instance, the M4Raw [4] in-distribution dataset intrinsically has lower SSIM scores due to noisy targets (low-field MRI, few coils), reducing average in-distribution performance. Excluding M4Raw increases average in-distribution performance from 0.952 to 0.959 for the unfiltered dataset.
>
> \
> Finally, we thank the reviewer for pointing out the typos. We will elaborate on the MVUE reference images in the revised version.
>
> \
> We hope our response and the additional results have clarified the reviewer’s concerns and questions. Since the reviewer found no major weaknesses in our work, we would be grateful if they would consider raising their score.
>
> \
> **References:**
>
> [1] S. Fu et al. “DreamSim: Learning New Dimensions of Human Visual Similarity using Synthetic Data.” NeurIPS, 2023.
>
> [2] C. Wang et al. “CMRxRecon: A publicly available 403 k-space dataset and benchmark to advance deep learning for cardiac MRI.” Scientific Data, 2024.
>
> [3] J. Zbontar et al. “fastMRI: An Open Dataset and Benchmarks for Accelerated MRI.” arXiv:1811.08839, 2019.
>
> [4] M. Lyu et al. “M4Raw: A multi-contrast, multi-repetition, multi-channel MRI k-space dataset for low-field.” Scientific Data, 2023.

---

> > ### Comment · Reviewer_vdoL · 2025-08-05
> >
> > I would like to thank the authors for their detailed responses. I especially appreciate the new performance break down in terms of anatomy, field strength and number of coils which I think is a good addition to the paper.
> >
> > I have read other reviews and authors' responses to them. Authors have addressed my concerns/questions successfully. Considering the responses and proposed changes, I will revise my rating to $5$ to recommend "accept".

---

> > > ### Author Response · Authors · 2025-08-06
> > >
> > > Thank you for the re-evaluation. We're glad that our response was helpful.

---

### Official Review · Reviewer_pcPX · 2025-06-30

**Rating:** 4
**Confidence:** 3

**Summary:**

This paper focuses on deep learning methods for accelerated MRI reconstruction, emphasizing the critical role of data quality and selection in improving model performance. The authors construct a large-scale training dataset comprising 18 publicly available k-space sources, and create a diverse evaluation suite of 48 test subsets, covering various anatomical structures, contrasts, magnet types, and coil configurations. The study systematically explores multiple data filtering strategies, including heuristic approaches and retrieval-based filtering using DreamSim similarity metrics, demonstrating their positive impact on the performance of state-of-the-art models.

**Dataset Code Accessibility:**

Yes

**Dataset Code Comments:**

The authors have made both the dataset and code publicly available, which greatly enhances the reproducibility and transparency of the research. Code URL: https://github.com/MLI-lab/data_filtering_for_accelerated_mri

**Ethical Considerations:**

No, there are no or only very minor ethics concerns

**Final Justification:**

The author addressed most of my concerns and I raised my rating to 4.

**Limitations Weaknesses:**

1. The workload appears somewhat limited — the authors primarily collect and re-organize public datasets, without proposing new models or substantial algorithmic innovations. The experiments are based solely on training and evaluating existing models.

2. The model used is relatively outdated — VarNet was introduced in 2020 and may not reflect the current state-of-the-art.

3. No qualitative comparison results are provided across different test sets, which could help assess the perceptual improvements claimed.

4. According to multiple figures, the SSIM improvements from data filtering are between 0.002 and 0.01. While statistically significant, the gains remain relatively modest.

5. The heuristic filtering methods (e.g., energy ratio threshold of 0.11 and edge density threshold of 0.017) are intuitive but lack rigorous justification or ablation. Although DreamSim-based filtering performs better, it mainly relies on statistical similarity and does not analyze the characteristics of retained vs. discarded images. The authors are encouraged to provide t-SNE or clustering visualizations to enhance interpretability of the filtering process.

**Strengths Contributions:**

1. While most existing work on MRI reconstruction focuses on architectural improvements, this paper addresses the novel and important question of how data impacts model performance, offering a fresh and valuable perspective.

2. The authors combine and utilize 18 publicly available k-space datasets, which significantly broadens the training and evaluation coverage.

---

> ### Author Rebuttal · Authors · 2025-07-31
>
> Thank you for reviewing our paper, and for noting that our work is “novel and important”, offers a “fresh and valuable perspective”, and for acknowledging our comprehensive experimental setup.
>
> In the following we address the concerns of the reviewer.
>
> 1. Regarding the concern that the workload is limited and we don’t propose new models and the statement that we don’t propose algorithmic innovations:
>
> Since we optimize the **training data**, we keep the models fixed, a setup common for data-centric works (e.g., DataComp [1,2]).
> \
> We introduce **novel** data curation methods, such as our **weighted retrieval filter**, which improves training data and enhances performance of existing models for accelerated MRI.
> \
> We also **unified** 18 diverse public k-space datasets, a non-trivial process detailed in our code repository, which will aid community access. Our work's results stem from analyzing over **200** models trained on this unified dataset.
>
> 2. Regarding the concern that VarNet [3] may not reflect the current state-of-the-art:
>
> VarNet still is the current **state-of-the-art** for end-to-end accelerated MRI. It is still winning recent competitions [4,5]. Since its introduction, models have improved very little in the past years for accelerated MRI, specifically on the fastMRI benchmark [6] (where the training data is fixed, and models are evaluated).
> \
> Nevertheless, to further provide evidence that our results are model agnostic, we evaluated model architectures that differ substantially from the VarNet. We evaluated plain U-nets [7] and Vision Transformers for accelerated MRI reconstruction [8] and found the same qualitative results as for VarNet. We will add these results to the revised version of the paper.
>
> 3. Regarding “no qualitative comparison results are provided across different test sets”:
>
> Qualitative comparisons across multiple test sets are provided in **Figure 1, 4, 13 and 14**. Those figures show visible improvements (sharper details and a reduction of artifacts) for weighted retrieval filtering over no filtering across multiple test sets.
>
> 4. Regarding the concern that the improvements, “while statistically significant, remain relatively modest”:
>
> While quantitative gains are indeed modest (also mentioned in the abstract), the gains obtained from our proposed filtering approach **visibly improve** the reconstruction, specifically we see sharper details and reduction of reconstruction artifacts (see Figure 1, 4 13 and 14).
>
> 5. Regarding the suggestion of an ablation study for hyperparameter choices of our proposed heuristic filtering methods, and the suggestion to provide a t-SNE plot for our DreamSim-based retrieval filtered datasets:
>
> For heuristic filtering we provide the following ablations:
> \
> Regarding energy threshold (Th) hyperparameter:
>
> |            | No filtering | Th = 0.11 (our choice) | Th = 0.35 |
> |----------|:--------------:|:-----------------------------:|:--------------:|
> |SSIM   | 0.952         | 0.953                           | 0.953        |
>
> We chose a threshold of 0.11 over 0.35, even though both yield the same performance. The 0.35 threshold resulted in the removal of many slices that contained clear signals, which defeated the purpose of the energy filter.
>
> Regarding edge-density threshold (Th) hyperparameter:
>
> |              | No filtering | Th = 0.017 (our choice) | Th = 0.025 |
> |----------|:--------------:|:-------------------------------:|:-------:|
> |SSIM   | 0.952         | 0.953                              | 0.952         |
>
> However, we note that while heuristic filtering methods were explored, we found the  DreamSim-based retrieval filtering to be more effective (see Section 3.3).
> \
> We also followed the suggestion of the reviewer and applied t-SNE on the DreamSim embeddings of the 120k slices dataset before and after retrieval filtering. We observed that the filtered dataset exhibits more distinct and evenly distributed clusters than the unfiltered dataset (unfortunately, we are not allowed to provide the plots during the rebuttal period).
> We will include the results in the revised version of the paper. We thank the reviewer for the suggestion.
>
> \
> We hope our response, including the additional results, has clarified the reviewer’s concerns. Given that the reviewer also finds our work novel and important, we would be happy if they would consider raising their score.
>
> \
> **References:**
>
> [1] S. Y. Gadre et al. “DataComp: In search of the next generation of multimodal datasets.” NeurIPS, 2023.
>
> [2] J. Li et al. “DataComp-LM: In search of the next generation of training sets for language models.”NeurIPS, 2024.
>
> [3] A. Sriram et al. “End-to-End Variational Networks for Accelerated MRI Reconstruction.” MICCAI, 2020.
>
> [4] J. Lyu et al. “The state-of-the-art in cardiac MRI reconstruction: Results of the CMRxRecon challenge in MICCAI 2023.” Medical Image Analysis, 2025.
>
> [5] F. Wang et al. “Towards Universal Learning-based Model for Cardiac Image Reconstruction: Summary of the CMRxRecon2024 Challenge.” aXiv:2503.03971, 2025.
>
> [6] M. J. Muckley et al. “Results of the 2020 fastMRI Challenge for Machine Learning MR Image Reconstruction.” IEEE Transactions on Medical Imaging, 2021.
>
> [7] O. Ronneberger et al. “U-Net: Convolutional Networks for Biomedical Image Segmentation”. MICCAI, 2015.
>
> [8] K. Lin et al. “Vision Transformers Enable Fast and Robust Accelerated MRI”. MIDL, 2022.

---

> > ### Comment · Reviewer_pcPX · 2025-08-04
> > **after rebuttal**
> >
> > Thank you for your reply, I have no further questions.

---

> > ### Comment · Reviewer_pcPX · 2025-08-05
> > **after rebuttal**
> >
> > I re-evaluate your manuscript in light of your clarifications and raise my overall score to a positive recommendation.

---

> > > ### Author Response · Authors · 2025-08-06
> > >
> > > Thank you for the re-evaluation. We're glad to know our response was helpful in clarifying your questions.

---

### Comment · Area_Chair_AN5X · 2025-08-04

Dear authors,

I am the AC here. Thanks for submitting the paper. I went through the review and noticed several review comments regarding the scope and efficiency of the filtering.

1. The workload appears somewhat limited — the authors primarily collect and re-organize public datasets, without proposing new models or substantial algorithmic innovations. The experiments are based solely on training and evaluating existing models.

2. The filtering process is computationally expensive, as it requires extracting DreamSim embeddings for all training and validation images.

I believe that this paper remains within the scope of the NeurIPS Benchmark and Dataset Track. This year, NeurIPS has revised its scope in this area.

**Data-centric AI methods and tools, e.g. to measure and improve data quality or utility, or studies in data-centric AI that bring important new insight.**

My question is about the filtering process. As a data-centric AI method, it clearly provided important new insights in terms of output. However, it came at the cost of high computational complexity. What is the rationale behind extracting DreamSim embeddings? Do operations at pixel level fail to achieve similar objectives?

---

> ### Author Response · Authors · 2025-08-06
>
> Dear AC,
>
> Thank you for handling our paper. We appreciate the opportunity to clarify the computational cost and our rationale for using DreamSim embeddings.
>
> The rationale behind using DreamSim embeddings is to leverage them for filtering so that the training data becomes better aligned with the test data. For this, we considered a filtering method that selects data that is similar to the validation data. After considering several image-based similarity  metrics, DreamSim appeared to be the most promising because the DreamSim paper [1] demonstrates that it works well compared to other common similarity metrics used for image retrieval.
> We're currently running an additional ablation study on comparing to a pixel-level similarity metric and hope to be able to share the result before the discussion phase ends.
>
> We also explored **pixel-level filtering methods**, which we refer to as heuristic filters (detailed in Section 3.1). While these filters can significantly reduce dataset size and offer a slight performance boost over no filtering, our findings in Section 3.3 show that DreamSim-based retrieval outperforms these heuristic methods. This is why we focus on DreamSim for our filtering approach.
>
> Finally, regarding the **computational complexity** of computing the embeddings: Computing DreamSim embeddings is cheap. It only requires a single forward pass of the dataset through the model. For example, on a single NVIDIA L40 GPU and a dataset size of 120k images (which we used in most of our experiments), computing the embeddings takes 0.6 hours. This is a minor computational cost compared to the 90 hours required to train the image reconstruction model.
>
> We hope this addresses your questions. Please let us know if there is anything else we can clarify.
>
> References:
>
> [1] S. Fu et al. “DreamSim: Learning New Dimensions of Human Visual Similarity using Synthetic Data.” NeurIPS, 2023.

---

> > ### Author Response · Authors · 2025-08-07
> > **Update on our ablation study: DreamSim vs. Pixel-Based Similarity**
> >
> > We've completed the ablation study comparing DreamSim and a pixel-based similarity metric for retrieval filtering. DreamSim outperforms the pixel-based approach.
> >
> > The pixel-level metric is computed as the Euclidean distance between two images in pixel-space. We ran the experiment on an unfiltered dataset of 40k slices for 8-fold accelerated MRI. The results are as follows:
> >
> > The numbers denote the average improvement in SSIM over no filtering on different test sets.
> >
> > |            | Cardiac | Non-cardiac| All test sets|
> > |----------|:--------------:|:------------:|:----:|
> > |Pixel-based | -0.01  | +0.003 | 0.0   |
> > |DreamSim | +0.006 | +0.006 | +0.006|
> >
> > We found that the pixel-based approach failed to retrieve a significant number of cardiac images, resulting in its poor performance on the cardiac test sets.

---

### Note · Authors · 2025-08-14

Dear AC,

Many thanks for handling our paper.

We appreciate the reviewers’ overall positive evaluation:
- “addresses the novel and important question of how data impacts model performance, offering a fresh and valuable perspective”,
- ”constructs a large-scale evaluation suite comprising 48 test sets, covering both in-distribution and out-of-distribution data, providing strong validation for generalization”,
- ”smaller training set (e.g., 40k slices), the filtered dataset outperforms much larger unfiltered datasets (e.g., 120k or 360k)”

We’d like to thank the AC and reviewers for engaging in the discussion. We are pleased that Reviewer pcPX and Reviewer vdoL both confirmed their concerns were addressed and that they raised their scores and recommend acceptance. We would like to use those final remarks to summarize our key clarifications.

**Clarification on our contribution and workload**

To clarify the nature of our contribution, our work introduces a novel data filtering algorithm as a data-centric method—an approach previously unexplored for MRI reconstruction. The validation of this method was a significant undertaking, requiring the standardization of 18 publicly available k-space sources and the training and analysis of over 200 models.

**Computational Cost of DreamSim**

The DreamSim filtering step is computationally cheap. Computing the DreamSim embeddings for 120k images only requires 0.6 hours, while the corresponding model training takes 90 hours. This cost is more than offset by the training compute saved, as filtering allows us to achieve superior results over using a larger but unfiltered dataset. Compared to pixel-level metrics, although pixel-level metrics are even faster, our ablation study shows that their performance is worse, which justifies the use of the perception metric DreamSim.

**Clarification on the method's practicality**

We leverage a validation set for filtering, which is small and often available, and has been leveraged in a similar form for other problems. Furthermore, our work also shows that even simpler heuristic filters, which do not rely on a validation set for retrieval, also yield performance gains.

---

### Decision · Program_Chairs · 2025-09-18

**Decision:**

Accept (poster)

**Comment:**

This paper focuses on deep learning methods for accelerated MRI reconstruction, emphasizing the critical role of data quality and selection in improving model performance. The authors construct a large-scale training dataset comprising 18 publicly available k-space sources, and create a diverse evaluation suite of 48 test subsets, covering various anatomical structures, contrasts, magnet types, and coil configurations. The study systematically explores multiple data filtering strategies, including heuristic approaches and retrieval-based filtering using DreamSim similarity metrics, demonstrating their positive impact on the performance of state-of-the-art models.

This paper has quite different natures than other regular papers in dataset and benchmark track, where this paper focused on

**Data-centric AI methods and tools, e.g. to measure and improve data quality or utility, or studies in data-centric AI that bring important new insight.**

Therefore specific considerations will be given when evaluating this paper. Therefore the core question is to evaluate whether to measure and improve data quality or utility, or studies in data-centric AI that bring important new insights.

I believe the answer is definitely yes. The reviewers have a consensus that

>  While most existing work on MRI reconstruction focuses on architectural improvements, this paper addresses the novel and important question of how data impacts model performance, offering a fresh and valuable perspective.
> The authors combine and utilize 18 publicly available k-space datasets, which significantly broadens the training and evaluation coverage.

I agree with their evaluations and believe this paper provides new insights in a practical problem in medical physics - MRI reconstruction.

I also noticed certain technical clarifications by authors such as operation on the pixel or embedding levels. Based on the additional experiments, I am satisfying their arguments.

Therefore the recommendation is acceptance.